# Blind Unlearning: Unlearning Without a Forget Set

## Abstract

Machine unlearning is the study of methods to efficiently remove the influence of some subset of the training data from the parameters of a previously-trained model. Existing methods typically require direct access to the "forget set" – the subset of training data to be forgotten by the model. This limitation impedes privacy, as organizations need to retain user data for the sake of unlearning when a request for deletion is made, rather than being able to delete it immediately. We first introduce the setting of *blind unlearning* – unlearning without explicit access to the forget set. Then, we propose a method for approximate unlearning called RELOAD, that leverages ideas from gradient-based unlearning and neural network sparsity to achieve blind unlearning. The method serially applies an ascent step with targeted parameter re-initialization and fine-tuning, and on empirical unlearning tasks, RELOAD often approximates the behaviour of a from-scratch retrained model better than approaches that leverage the forget set. Finally, we extend the blind unlearning setting to *blind remedial learning*, the task of efficiently updating a previously-trained model to an amended dataset[1].

## 1 Introduction

Machine unlearning poses the problem of removing the influence of certain instances in the training data on a given statistical model (Bourtoule et al., 2019). Motivated by "right to be forgotten" provisions, like those in the European Union's General Data Protection Regulation (GDPR) (European Parliament & Council of the European Union), methods in machine unlearning aim to provide efficient means to "forget" specific data points from a trained model without requiring that it be retrained from scratch. As larger models become more prevalent (Achiam et al., 2023), the need to unlearn specific data instances without retraining from scratch is increasingly important.

Contemporary unlearning methods generally require explicit access to the so-called "forget set" – the subset of training data to be forgotten by the model. For example, one approach entails performing steps of gradient ascent on the loss landscape characterized by the forget set in order to remove its influence on the model weights (Thudi et al., 2022). However, the reliance of these methods on the forget set introduces a tension in the context of preserving user privacy: in order to enable unlearning, organizations must retain the complete original set of user data on which the model was trained. The retention of this data, even for the purpose of unlearning, can expose organizations and individuals to risks associated with data breaches or unauthorized access. To bridge this gap, there is a clear need for unlearning methods that operate without requiring access to the forget set. Existing work aims to reduce the reliance on the forget set, but is limited to the constrained task of forgetting classes of data, and requires knowing which classes are being forgotten (Tarun et al., 2023).

This work presents an algorithm for machine unlearning in the absence of an explicitly defined forget set; a setting we establish as "blind unlearning." Our method, RELOAD, assumes that the modeller only has access to (a) a model trained on a dataset $\mathcal{D}$, (b) the "retain set," $\mathcal{D}_{new} \triangleq \mathcal{D} \backslash \mathcal{D}_{forget}$, and (c) cached gradients from the last iteration of training on $\mathcal{D}$. Notable in its absence from these requirements is the forget set – this means that RELOAD allows for deletion of instances in $\mathcal{D}_{forget}$ at the conclusion of training without inhibiting downstream unlearning.

In this vein, our work makes the following contributions:

---

[1] A software implementation of our work can be found in this code repository.

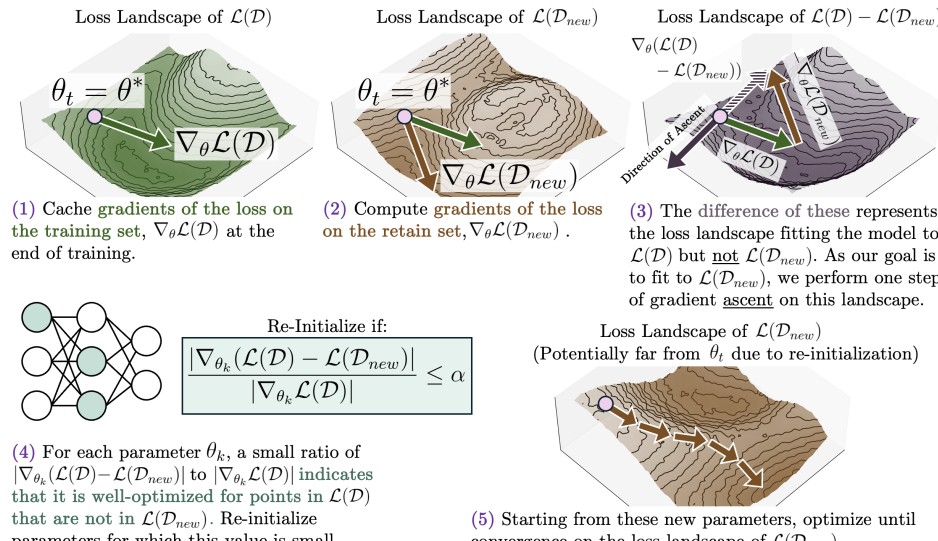

(1) Cache gradients of the loss on the training set, $\nabla_\theta \mathcal{L}(\mathcal{D})$ at the end of training.

(2) Compute gradients of the loss on the retain set, $\nabla_\theta \mathcal{L}(\mathcal{D}_{new})$ .

(3) The difference of these represents the loss landscape fitting the model to $\mathcal{L}(\mathcal{D})$ but not $\mathcal{L}(\mathcal{D}_{new})$. As our goal is to fit to $\mathcal{L}(\mathcal{D}_{new})$, we perform one step of gradient ascent on this landscape.

Re-Initialize if:
$$\frac{|\nabla_{\theta_k}(\mathcal{L}(\mathcal{D}) - \mathcal{L}(\mathcal{D}_{new}))|}{|\nabla_{\theta_k}\mathcal{L}(\mathcal{D})|} \le \alpha$$

(4) For each parameter $\theta_k$, a small ratio of $|\nabla_{\theta_k}(\mathcal{L}(\mathcal{D}) - \mathcal{L}(\mathcal{D}_{new})|$ to $|\nabla_{\theta_k}\mathcal{L}(\mathcal{D})|$ indicates that it is well-optimized for points in $\mathcal{L}(\mathcal{D})$ that are not in $\mathcal{L}(\mathcal{D}_{new})$. Re-initialize parameters for which this value is small.

(5) Starting from these new parameters, optimize until convergence on the loss landscape of $\mathcal{L}(\mathcal{D}_{new})$.

Figure 1: High-level overview of the RELOAD algorithm for blind approximate unlearning and remedial learning. The algorithm marries a gradient-based unlearning step modified for the blind unlearning setting (Steps (1) through (3)) with a weight saliency-based selective re-initialization (Step (4)) and subsequent fine-tuning (Step (5)). Because the blind unlearning setting prohibits taking gradients with respect to $\mathcal{D}_{forget}$, RELOAD exploits the linearity of differentiation to treat $\nabla_\theta(\mathcal{L}(\mathcal{D}) - \mathcal{L}(\mathcal{D}_{new}))$ as a proxy for $\nabla_\theta\mathcal{L}(\mathcal{D}_{forget})$ at the location in parameter space corresponding to $\theta_t$. This allows us to apply one gradient step in this direction. Intuitively, this update in Step (3) removes information about $\mathcal{D}_{forget}$ from *all* network parameters, while the re-initialization in Step (4) re-initialises those parameters with a uniquely strong correspondence to $\mathcal{D}_{forget}$ (for which a single ascent step will not fully remove this information). RELOAD achieves state-of-the-art performance on a collection of unlearning tasks, often outperforming baselines with direct access to $\mathcal{D}_{forget}$.

1. We introduce RELOAD, an algorithm for approximate blind unlearning in parametric models. RELOAD marries ideas from gradient-based unlearning algorithms and neural network sparsity to achieve blind unlearning. We formally show that the requirements of the RELOAD algorithm satisfy the blind unlearning by not permitting recoverability of instances in the forget set in the common setting of softmax classification.

2. Empirical evaluations demonstrate that RELOAD effectively tackles machine unlearning in several diverse settings, often faithfully approximating the behaviour of a from-scratch retrained model *better than existing unlearning approaches that leverage the forget set*.

3. We extend the RELOAD framework to the setting of "remedial learning," which aims to efficiently update a statistical model given that some instances in the training data that have been amended since the model was originally trained. This enables computationally-efficient data correction in pretrained models without the need for costly retraining.

## 2 BACKGROUND

### 2.1 SETTING AND NOTATION

Let $\mathcal{D} = \{(X_i, Y_i)\}_{i=1,...,N}$ represent a collection of i.i.d. data, where $X \in \mathcal{X}$ represents input covariates and $Y \in \mathcal{Y}$ represents labels for supervised learning. Then, for some class of models $\mathcal{M}$, let $\theta^*$ represent the parameters that minimize the empirical loss with respect to training data $\mathcal{D}$,

$$\theta^* \triangleq \underset{\theta \in \Theta}{\arg\min} \; \mathbb{E}_{(X_i,Y_i) \sim \mathcal{D}} \; \mathcal{L}((X_i, Y_i); \theta). \tag{1}$$

We denote an instantiation of $\mathcal{M}$ trained on $\mathcal{D}$ as $\mathcal{M}^{(\theta^*)}$. After $\mathcal{M}^{(\theta^*)}$ is trained, assume that some transformation is applied to $\mathcal{D}$ to yield $\mathcal{D}_{new}$ (e.g., deleting instances from $\mathcal{D}$ that are in $\mathcal{D}_{forget}$). Then, $\theta^\sim$ represents the parameters that minimize the empirical loss with respect to $\mathcal{D}_{new}$,

$$\theta^\sim \triangleq \underset{\theta \in \Theta}{\arg\min}\, \mathbb{E}_{(X_i, Y_i) \sim \mathcal{D}_{new}}\; \mathcal{L}((X_i, Y_i); \theta). \tag{2}$$

Our general goal, encompassing both unlearning and remedial learning, is transforming $\mathcal{M}^{(\theta^*)}$ into $\mathcal{M}^{(\theta^\sim)}$ without naively obtaining $\mathcal{M}^{(\theta^\sim)}$ by re-training an instance of $\mathcal{M}$ on $\mathcal{D}_{new}$ from scratch.

**Machine Unlearning.** In the machine unlearning setting, the transformation of $\mathcal{D}$ into $\mathcal{D}_{new}$ consists of first identifying a subset of the data whose influence to remove, $\mathcal{D}_{forget}$, and taking $\mathcal{D}_{new} \triangleq \mathcal{D} \backslash \mathcal{D}_{forget}$. These remaining instances represent the portion of the training data that is retained – the full training set, less those instances to be forgotten. The goal of approximate unlearning methods – of which RELOAD is one (see Section 2.2) – is to efficiently learn an approximation of $\mathcal{M}^{(\theta^\sim)}$. The classical setting assumes that the modeller has access to the trained model $\mathcal{M}^{(\theta^*)}$, the training dataset $\mathcal{D}$, the remaining data $\mathcal{D}_{new}$, and the forget set $\mathcal{D}_{forget}$ (Cao & Yang, 2015).

**Remedial Learning.** The unlearning setting is subject to the restriction that $\mathcal{D}_{new} \triangleq \mathcal{D} \backslash \mathcal{D}_{forget}$; however, one may also consider the broader setting wherein $\mathcal{D}_{new}$ is the result of some arbitrary item-wise transformation to $\mathcal{D}$. Formally, let $f : \mathcal{X} \times \mathcal{Y} \to \mathcal{X} \times \mathcal{Y}$ denote a transformation, and write $(X'_i, Y'_i) = f(X_i, Y_i)$. Then, $\mathcal{D}_{new}$ represents the result of applying $f$ item-wise to $K$ elements of $\mathcal{D}$, and applying the identity transform to the remaining $N - K$ elements, as

$$\mathcal{D}_{new} = \{(X'_i, Y'_i)\}_{i=1,...,K} \cup \{(X_i, Y_i)\}_{i=K+1,...,N}. \tag{3}$$

This represents a generalization of the unlearning problem, as we wish to "unlearn" the influence of $\{(X_i, Y_i)\}_{i=1,...,K}$ on our original model, and "relearn" the influence of $\{(X'_i, Y'_i)\}_{i=1,...,K}$. This setting encompasses the following data transformations, among others:

1. *Covariate Correction*: $\mathcal{D}_{new} = \{(X'_i, Y_i)\}_{i=1,...,K} \cup \{(X_i, Y_i)\}_{i=K+1,...,N}$, where $X'_i$ represents a corrected version of the features $X_i$, and indices $K + 1, ..., N$ correspond to those with erroneous covariates (e.g., data was corrupted during collection/pre-processing).

2. *Label Correction*: $\mathcal{D}_{new} = \{(X_i, Y'_i)\}_{i=1,...,K} \cup \{(X_i, Y_i)\}_{i=K+1,...,N}$, where $Y'_i$ represents a corrected version of the label $Y_i$, and indices $K+1, ..., N$ correspond to those that were originally mis-labelled during annotation.

3. *Backdoor Removal*: $\mathcal{D}_{new} = \{(X'_i, Y_i)\}_{i=1,...,K} \cup \{(X_i, Y_i)\}_{i=K+1,...,N}$, where $X'_i$ represents a version of the features $X_i$ lacking the injected backdoor pattern, and indices $K + 1, ..., N$ correspond to those that were originally transformed with a backdoor during processing. Models trained with backdoors in the training set learn shortcuts (Geirhos et al., 2020), which can be exploited to induce misclassification.

This work studies how the RELOAD algorithm accomplishes tasks both in the unlearning setting, and in the setting of remedial learning.

**Blind Unlearning / Blind Remedial Learning.** In contrast to the classical unlearning (and remedial learning) settings, in which the modeller has access to the forget set, our setting assumes no such access. We call this setting *blind unlearning* (or *blind remedial learning*). The blind unlearning / remedial learning setting has access to the trained model $\mathcal{M}_{\mathcal{D}}$, the new dataset $\mathcal{D}_{new}$, and (potentially) some limited information about the original data, $\mathcal{I}_{\mathcal{D}}$, from which which neither $\mathcal{D}_{forget}$ (in the unlearning setting) or $\mathcal{D}$ (in the remedial learning setting) can be fully reconstructed.

## 2.2 RELATED WORK

**Exact and Approximate Unlearning.** Exact unlearning refers to the subclass of algorithms that provide formal guarantees of the extent to which information about $\mathcal{D}_{forget}$ was removed from the weights of a model. The trivial method for exact unlearning consists of naively retraining the model from scratch (this is considered the gold-standard for machine unlearning; see Cao & Yang (2015); Thudi et al. (2022); Shaik et al. (2024)). Other exact unlearning methods include SISA (Bourtoule et al., 2019), which partitions the data to accelerate retraining, Certified Data Removal

(Guo et al., 2019), which performs a Newton update in the opposite direction of the gradient with respect to $\mathcal{D}_{forget}$, and Certified Graph Unlearning (Chien et al., 2022), which builds on Certified Data Removal using the graph topology to enforce guaranteed unlearning. Unlike exact unlearning methods, approximate unlearning algorithms (like RELOAD) aim to recover the behaviour of a model naively retrained on the new set without providing any formal theoretical guarantees. These methods can be sub-classified into either gradient-based or weight-saliency based algorithms.

**Gradient-Based Approximate Unlearning.** Many existing approximate unlearning algorithms perform an optimization procedure on $\mathcal{M}^{(\theta^*)}$ using the forget set $\mathcal{D}_{forget}$ and the retain set $\mathcal{D}_{new}$. One simple standard approach applies gradient ascent on the loss with respect to $\mathcal{D}_{forget}$, in order to undo the parameter updates induced by those instances during training (Graves et al., 2021; Thudi et al., 2022). Another gradient-based approach leverages a teacher-student method: "Bad Teacher" performs knowledge distillation based on one trained model on $\mathcal{D}_{new}$ (the "good teacher") and one a randomly initialised model on $\mathcal{D}_{forget}$ (the "bad teacher") (Chundawat et al., 2022); SCalable Remembering and Unlearning unBound (SCRUB) similarly distills a student model from a teacher trained on $\mathcal{D}$, but the student learns to selectively disobey the teacher by directly maximizing the loss on $\mathcal{D}_{forget}$ (Kurmanji et al., 2023). A third family directly manipulates the structure of the learned representation space using gradients: Distance-based Unlearning via Centroid Kinematics (DUCK) (Cotogni et al., 2023) drives representations of elements in $\mathcal{D}_{forget}$ towards the nearest incorrect centroid in the feature space, while Boundary Unlearning (Chen et al., 2023) implements class-level unlearning by shifting the decision boundary corresponding to the class(es) defining $\mathcal{D}_{forget}$.

**Weight Saliency-Based Approximate Unlearning.** Another class of approximate unlearning methods derives from the hypothesis that identifiable substructures in neural networks often correspond to different subsets of the training data (Pfeiffer et al., 2023). These methods leverage ideas from neural sparsity (Frankle & Carbin, 2018; Chen et al., 2024) to perform targeted unlearning on specific parameters. Saliency unlearning (SalUn) uses a threshold on $\nabla_\theta \mathcal{L}(\mathcal{D}_{forget})$ to identify parameters containing the most signal about $\mathcal{D}_{forget}$ and focuses model updates on these parameters (Fan et al., 2023). Selective Synaptic Dampening (SSD) (Foster et al., 2023) extends this idea to avoid gradient steps by scaling parameters based on their Fisher Information Matrix importance scores.

**Blind Unlearning.** This setting involves unlearning without access to $\mathcal{D}_{forget}$ at the instanced of unlearning. It then reduces to taking a model fit on one dataset $\mathcal{D}$ and adapting it to fitting a new dataset $\mathcal{D}_{new}$. This connects to domain adaptation, in which differences in datasets may not be explicitly defined. In blind unlearning it is not available. An unlearning baseline, Finetuning (FT) (Warnecke et al., 2023) on the retain-set $\mathcal{D}_{new}$ fulfills the blind criteria. Catastrophically forgetting the last $k$ layers (CF-$k$) and Exact-unlearning the last $k$ layers (EU-$k$) (Goel et al., 2022) are also blind. Fisher Forgetting (Fisher) (Golatkar et al., 2020) is also a blind unlearning algorithm, but is theoretically bound by class unlearning. Both FT and CF-$k$ provide no strong theoretical indication of unlearning. EU-$k$ does by re-initialising the last k layers of the model. Our method, RELOAD, provides a stronger indication by selectively re-initialises parameters which know the most about the knowledge we wish to remove.

## 3 RELOAD

### 3.1 ALGORITHM REQUIREMENTS

**Assumption 1** (Unlearning from Cached Gradients). *In* RELOAD, $\mathcal{I}_\mathcal{D} = \nabla_\theta \mathcal{L}(\mathcal{D})$.

The following lemma demonstrates why this is a valid choice of $\mathcal{I}_\mathcal{D}$ for blind unlearning / remedial learning in the softmax classification setting, because this choice does not not permit recovery of the removed instances within $\mathcal{D}_{forget}$ (or of instances in $\mathcal{D}$ and not in $\mathcal{D}_{new}$ in remedial learning) in the common setting of softmax classification.

**Definition 1** (Recoverability). *Consider some data, $\mathcal{D} \in \mathscr{D}$, and consider a transformation $f : \mathscr{D} \to \mathcal{Q}$ that maps $\mathcal{D}$ into an arbitrary output space $\mathcal{Q}$. $\mathcal{D}$ is* recoverable *if $f$ is injective.*

**Lemma 1** ($\mathcal{D}_{forget}$ is Not Recoverable from $\mathcal{I}_\mathcal{D}$ in Softmax Classification). *Consider softmax classification over $C$ classes, where each $Y_i \in [0,1]^C$ represents a one-hot encoded vector of class labels, $\hat{Y}_i = \left[ e^{Z_{i1}} / \sum_{j=1}^C e^{Z_{j1}}, ..., e^{Z_{iC}} / \sum_{j=1}^C e^{Z_{jC}} \right]$ represents predicted probabilities for each class*

generated from model logits $Z_i \in \mathbb{R}^C$, and $\mathcal{L}((X_i, Y_i), \hat{Y}_i) = -\sum_{i=1}^N \sum_{j=1}^C Y_{ij} \log \hat{Y}_{ij}$. The transformation $\mathcal{G} : \mathscr{D} \to \Theta$ s.t. $\mathcal{G}(\mathcal{D}) = \nabla_\theta \mathcal{L}(\mathcal{D}) \triangleq \mathcal{I}_\mathcal{D}$ is not injective.

*Proof.* Recall from Section 3.2 that we can write $\nabla_\theta \mathcal{L}(\mathcal{D}) - \nabla_\theta \mathcal{L}(\mathcal{D}_{new})$ as $\nabla_\theta \mathcal{L}(\mathcal{D}_{forget})$. Then, if $|\mathcal{D}_{forget}| > 1$, the numerator $\nabla_{\theta_k} \mathcal{L}(\mathcal{D}_{forget})$ can be written as $\sum_{(X_i, Y_i) \in \mathcal{D}_{forget}} \nabla_{\theta_k} \mathcal{L}((X_i, Y_i), \hat{Y}_i)$. Because summation is not injective, $\mathcal{G}$ is also non-injective. In the other case, if $|\mathcal{D}_{forget}| = 1$, we write $\mathcal{D}_{forget} = \{(X_1, Y_1)\}$. Without loss of generality, $Y_{ij} = 1$ and $Y_{ik} = 0$ for all $k \neq j$. Given $\nabla_{\theta_k} \mathcal{L}((X_1, Y_1), \hat{Y}_1) = -\nabla_{\theta_k} \sum_{i=1}^C Y_{1i} \log \hat{Y}_{1i} = \nabla_{\theta_k} \log \hat{Y}_{1j} = \frac{1}{\hat{Y}_{1j}}$, we can recover the $j$'th output of the model, $\hat{Y}_{1j}$. $\hat{Y}_{1j} = \frac{e^{Z_{1j}}}{\sum_{i=1}^C e^{Z_{1i}}}$. For any element $Z_{1k}$ of $Z_1$, $e^{Z_{1k}} = \hat{Y}_{1j} \cdot \sum_{i=1}^C e^{Z_{1i}}$, this implies that $Z_{1k} = \log(\hat{Y}_{1j} \cdot \sum_{i=1}^C e^{Z_{1i}}) = \log(\hat{Y}_{1j}) + \log(\sum_{i=1}^C e^{Z_{1i}}))$ which cannot be calculated without knowing the other elements of $Z_1$. Thus, given only $\hat{Y}_{1j}$, no elements of $Z_1$ can be obtained, hence $\mathcal{G}$ is also injective in this case. $\square$

## 3.2 ALGORITHM INTUITION

**Direction of Movement.** The central challenge of blind unlearning is that taking repeated gradients of $\mathcal{L}(\mathcal{D}_{forget})$ is impossible without access to $\mathcal{D}_{forget}$. However, from cached gradients of $\mathcal{D}$ at the conclusion of model training, $\nabla_\theta \mathcal{L}(\mathcal{D})$, we can infer $\nabla_\theta \mathcal{L}(\mathcal{D}_{forget})$.

To do so, let $\hat{Y}_i = \mathcal{M}^{(\theta)}(X_i)$ represent the model's prediction. Then,

$$\nabla_\theta \mathcal{L}(\mathcal{D}_{forget}) = \sum_{(X_i, Y_i) \in \mathcal{D}_{forget}} \nabla_\theta \mathcal{L}((X_i, Y_i), \hat{Y}_i) = \sum_{(X_i, Y_i) \in \mathcal{D} \backslash \mathcal{D}_{new}} \nabla_\theta \mathcal{L}((X_i, Y_i), \hat{Y}_i) \quad (4)$$

where the second equality follows from $\mathcal{D}_{new} = \mathcal{D} \backslash \mathcal{D}_{forget}$. Equivalently,

$$= \sum_{(X_i, Y_i) \in \mathcal{D}} \nabla_\theta \mathcal{L}((X_i, Y_i), \hat{Y}_i) - \mathbb{1}_{(X_i, Y_i) \in \mathcal{D}_{new}} \left[ \nabla_\theta \mathcal{L}((X_i, Y_i), \hat{Y}_i) \right] \quad (5)$$

$$= \sum_{(X_i, Y_i) \in \mathcal{D}} \nabla_\theta \mathcal{L}((X_i, Y_i), \hat{Y}_i) - \sum_{(X_i, Y_i) \in \mathcal{D}_{new}} \nabla_\theta \mathcal{L}((X_i, Y_i), \hat{Y}_i) \quad (6)$$

$$= \nabla_\theta \mathcal{L}(\mathcal{D}) - \nabla_\theta \mathcal{L}(\mathcal{D}_{new}). \quad (7)$$

Therefore, a gradient-based descent update in the direction of $\nabla_\theta \mathcal{L}(\mathcal{D}_{forget})$ moves the model parameters such that they better fit to $\mathcal{D}_{forget}$; because our goal is *unlearning* $\mathcal{D}_{forget}$, RELOAD instead begins with a single gradient *ascent* update step in this direction.

In unlearning, our goal is to obtain a gradient in the direction of $\mathcal{D}_{forget}$. The remedial learning case is more general: the goal is to obtain $\nabla_\theta \mathcal{L}(\mathcal{D} \cap \mathcal{D}_{new}^c)$, a gradient pointing towards the empirical minimum of the loss on elements that are uniquely contained in $\mathcal{D}$ and not in $\mathcal{D}_{new}$, and $-\nabla_\theta \mathcal{L}(\mathcal{D}^c \cap \mathcal{D}_{new})$, a gradient pointing *away* from the empirical minimum of the loss on elements uniquely contained in $\mathcal{D}_{new}$ but not in $\mathcal{D}$. Unlearning represents the special case of this framework in which $\mathcal{D} \cap \mathcal{D}_{new}^c = \mathcal{D}_{forget}$ and $\mathcal{D}^c \cap \mathcal{D}_{new} = \emptyset$. In the remedial learning setting, the desired gradient is also $\nabla_\theta \mathcal{L}(\mathcal{D}) - \nabla_\theta \mathcal{L}(\mathcal{D}_{new})$; the derivation can be found in Appendix A.1. This informs Step (2 - 3) in Figure 1.

**Targeted Parameter Adjustments.** Taking a gradient step in this direction, however, is insufficient for unlearning (or remedial learning) for two reasons. First, we are limited to a single gradient step in this direction (Assumption 1), and second, theory from network modularity (Rodriguez et al., 2019) suggests that a small subset of parameters contain a disproportionate amount of the necessary information to characterize instances in $\mathcal{D}_{forget}$. While one ascent step may be useful at removing what little information about $\mathcal{D}_{forget}$ is included *across all* network parameters, it is less plausible that this single step will remove information about $\mathcal{D}_{forget}$ from the subset of parameters most responsible for its characterization.

We therefore perform selective re-initialization of these parameters as follows. Consider the gradient $\nabla_{\theta_k} \mathcal{L}(\mathcal{D}_{forget})$, the gradient of the loss with respect to instances in $\mathcal{D}_{forget}$ and with respect to a

particular parameter $\theta_k$. If this gradient is small, it means that $\theta_k$ is well-optimized to characterize instances in $\mathcal{D}_{forget}$; if this gradient is large, it means that $\theta_k$ poorly characterizes these instances. Although the absolute values of these gradients are largely meaningless, the relative magnitude of $\nabla_{\theta_k}\mathcal{L}(\mathcal{D}_{forget})$ compared $\nabla_{\theta_k}\mathcal{L}(\mathcal{D})$ is a meaningful representation of the extent to which $\theta_k$ is responsible for characterizing information about $\mathcal{D}_{forget}$. We call this the *knowledge value* of parameter $\theta_k$, and formally define it as,

$$KV_{\theta_k} \triangleq \frac{|\nabla_{\theta_k}\mathcal{L}(\mathcal{D}_{forget})| + \epsilon}{|\nabla_{\theta_k}\mathcal{L}(\mathcal{D})| + \epsilon} = \frac{|\nabla_{\theta_k}\mathcal{L}(\mathcal{D}) - \nabla_{\theta_k}\mathcal{L}(\mathcal{D}_{new})| + \epsilon}{|\nabla_{\theta_k}\mathcal{L}(\mathcal{D})| + \epsilon}, \qquad (8)$$

where $\epsilon$ is a small Laplace smoothing constant. Here the second equality follows from the relationship between $\nabla_{\theta_k}\mathcal{L}(\mathcal{D}_{forget})$, $\nabla_{\theta_k}\mathcal{L}(\mathcal{D})$, and $\nabla_{\theta_k}\mathcal{L}(\mathcal{D}_{new})$ that we derived earlier in this section. A small knowledge value characterizes a parameter that is knowledgeable about $\mathcal{D}_{forget}$, so by selectively re-initializing all parameters $\theta_k$ if $\text{QUANTILE}_{KV}(KV_{\theta_k}) \leq \alpha$ ($\alpha$ is a hyperparameter), we can remove the influence of the parameters uniquely responsible for encoding information about these data. This thinking extends on lines of work in gradient-based input saliency maps (Smilkov et al., 2017) and saliency unlearning by Fan et al. (2023). We explore and compare other methods of identifying knowledgeable weights in Appendix A.5.2. This informs Step (4) in Figure 1.

### 3.3 THE RELOAD ALGORITHM

Based on this intuition, our RELOAD algorithm contains the following steps. (1) Cache the gradients $\nabla_\theta\mathcal{L}(\mathcal{D})$ at the end of training. (2) Compute $\nabla_\theta\mathcal{L}(\mathcal{D}_{new})$. (3) Perform one step of gradient *ascent* in the direction of $\nabla_\theta(\mathcal{L}(\mathcal{D}) - \mathcal{L}(\mathcal{D}_{new}))$. (4) Re-initialize all parameters $\theta_k$ that are smaller than the $\alpha$-QUANTILE of knowledge values. Finally, (5) fine-tune until convergence on $\mathcal{L}(\mathcal{D}_{new})$. A formal description is shown in Algorithm 1. A software implementation can be found here.

---

**Algorithm 1** The RELOAD Algorithm for Blind Unlearning and Remedial Learning

---

1: **Input:** $\mathcal{M}^{(\theta^*)}$, cached $\nabla_\theta\mathcal{L}(\mathcal{D})$, $\mathcal{D}_{new}$
2: **Parameters:** $\eta_p$: priming step learning rate, $\epsilon$: noise parameter, $\alpha$: reset proportion
3: **Output:** Trained model approximating $\mathcal{M}^{(\theta^\sim)}$
4:
5: **procedure** RELOAD($\mathcal{M}^{(\theta^*)}$, $\nabla_\theta\mathcal{L}(\mathcal{D}; \mathcal{M}^{(\theta^*)})$, $\mathcal{D}_{new}$)
6: $\quad \theta' \leftarrow \theta^* + \eta_p \nabla_\theta(\mathcal{L}(\mathcal{D}) - \mathcal{L}(\mathcal{D}_{new}))$ $\qquad\qquad\qquad$ ▷ Step (2 − 3) *(Fig. 1)*
7: $\quad \text{KV} \leftarrow \left\{ \frac{|\nabla_{\theta_k}\mathcal{L}(\mathcal{D}) - \nabla_{\theta_k}\mathcal{L}(\mathcal{D}_{new})| + \epsilon}{|\nabla_{\theta_k}\mathcal{L}(\mathcal{D})| + \epsilon} \right\}_{\theta_k \in \theta}$ $\qquad$ ▷ Step (3) *(Fig. 1)*
8: $\quad$ **for** $\theta_k \in \theta'$ **do**
9: $\qquad$ **if** $\text{QUANTILE}_{KV}(KV_{\theta_k}) \leq \alpha$ **then**
10: $\qquad\quad \theta'_k \leftarrow \text{INITIALIZE}(\cdot)$ $\qquad\qquad\qquad\qquad\qquad\qquad$ ▷ Step (4) *(Fig. 1)*
11: $\qquad$ **end if**
12: $\quad$ **end for**
13: $\quad$ Train $\mathcal{M}^{(\theta')}$ to convergence on $\mathcal{D}_{new}$ $\qquad\qquad\qquad\qquad$ ▷ Step (5) *(Fig. 1)*
14: **end procedure**

---

## 4 RESULTS AND ANALYSIS

### 4.1 METHODOLOGICAL INTROSPECTION

Figure 2 introspects on the selected feature maps of a ResNet-18 model when using RELOAD to unlearn the class "8" from the SVHN dataset. The experiment demonstrates the importance of the re-initialization step (Step (4) in Figure 1), as even after a single ascent step, the model still finds "8" to be the most probable class. It is only after the salient weights are identified and re-initialized that the model emits a lower-entropy distribution classifying the digit as a "2". This suggests that the primary utility of the ascent step in our algorithm is in amending the representations of $\mathcal{D}_{forget}$ in the later layers of the network, while the salient weight re-initialization updates also modify the representations produced by earlier layers. The findings of this experiment present empirical confirmation of the intuition used to develop the algorithm (Section 3.2).

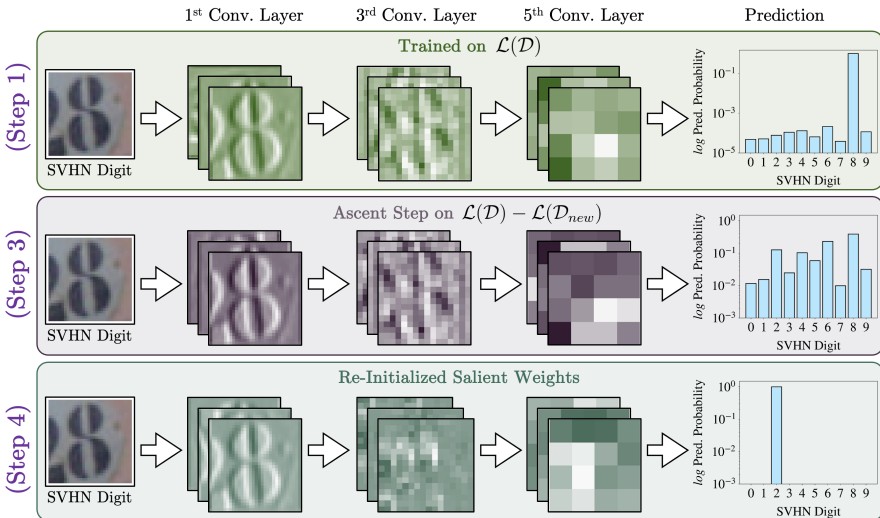

Figure 2: Introspecting on selected feature maps of a ResNet-18 model when using RELOAD to unlearn the class "8". For brevity, we selected the first channel from each feature map for the sake of visualization, though the patterns we identify appear to hold more broadly across channels. (*Top*) The feature maps (activations) of the first, third, and fifth convolutional layers after the model was initially trained on $\mathcal{D}$ (Step (**1**) in Figure 1). (*Middle*) These same feature masks after the ascent step has been applied (Step (**3**) in Figure 1). Observe how the activations of the model remain largely unchanged, although the logits represent a considerably more uniform distribution over the digits. (*Bottom*) These same features masks after the salient weights have been identified and re-initialized (Step (**4**) in Figure 1). Observe that the activation of the first convolutional layer is largely unchanged – this is expected, as the earlier layers of the network correspond to broad feature detectors (Zeiler & Fergus, 2014)) that may be less unique to the features of any particular class in this data. Notice, however, that the feature map of the third convolutional layer is substantially different from that of the previous two stages (it no longer features a hazy "8"), and that the network now emits a significantly lower-entropy distribution predicting the input image as a "2".

## 4.2 UNLEARNING EXPERIMENTS

**Baselines.** We compare RELOAD against baseline approaches of gradient ascent (GA) (Thudi et al., 2022), fine-tuning on $\mathcal{D}_{new}$ (FT) (Warnecke et al., 2023), Selective Synaptic Dampening (SSD) (Foster et al., 2023), SCalable Remembering and Unlearning unBound (SCRUB) (Kurmanji et al., 2023), Catastrophically Forgetting the last $k$ layers (CF-$k$) (Goel et al., 2022), Exact-Unlearning the last $k$ layers (EU-$k$) (Goel et al., 2022), SalUn (Fan et al., 2023), and Fisher forgetting (Fisher) (Golatkar et al., 2020). Of these baselines, the requirements of FT, CF-$k$, EU-$k$, and Fisher satisfy the blind unlearning setting, whereas the others require direct access to $\mathcal{D}_{forget}$.

**Evaluation.** As the goal of approximate unlearning is to produce a learned model that mimics the behaviour of $\mathcal{M}^{(\theta^{\sim})}$, we employ several evaluation statistics to measure the similarity in performance between our learned model and a version of $\mathcal{M}^{(\theta^{\sim})}$ that we naively train from scratch. The accuracy on $\mathcal{D}_{new}$ (NA, ↑) measures how well each learned model fits to the new data. The difference in accuracy on $\mathcal{D}_{forget}$ (ΔFA, ↓) measures the difference in accuracy between our learned model and $\mathcal{M}^{(\theta^{\sim})}$ on $\mathcal{D}_{forget}$, while the difference in error on $\mathcal{D}_{forget}$ (ΔFE, ↓) measures the difference in (cross-entropy) loss between our learned model and $\mathcal{M}^{(\theta^{\sim})}$ on the $\mathcal{D}_{forget}$. The difference in success rates of a membership inference attack on $\mathcal{D}_{forget}$ (ΔFMIA, ↓) measures the ability of the inference attack from Shokri et al. (2017) to identify members of $\mathcal{D}_{forget}$ in the training data of each leaned model, compared to the baseline rate of identification on $\mathcal{M}^{(\theta^{\sim})}$. We also report the AUC of the MIA attack model (ΔAUC, ↓). The symmetric KL-divergence on $\mathcal{D}_{new}$ (NSKL, ↓) measures the dissimilarity in the logits produced by our learned model and $\mathcal{M}^{(\theta^{\sim})}$ on $\mathcal{D}_{new}$, while the symmetric KL-divergence on $\mathcal{D}_{forget}$ (FSKL, ↓) measures the dissimilarity in the logits produced by our learned model and $\mathcal{M}^{(\theta^{\sim})}$ on $\mathcal{D}_{forget}$. Cost (↓) measures the computational cost of the method, and is the ratio of time to run the method to the time to naively train $\mathcal{M}^{(\theta^{\sim})}$.

**RELOAD unlearns randomly-selected samples.** In this experiment, we randomly assign 10% of the training data samples to $\mathcal{D}_{forget}$, to showcase how well each method can unlearn arbitrary training samples. The results of this experiment are shown in Table 1. Observe that RELOAD achieves the highest NA, while maintaining the lowest $\Delta$FA, $\Delta$FE, $\Delta$FMIA, and FSKL of all approaches. This suggests that RELOAD successfully approximates $\mathcal{M}^{(\theta^{\sim})}$ better than the baselines. That fine-tuning achieves a lower NSKL than RELOAD is hardly surprising, as NSKL measures dissimilarity in logits on $\mathcal{D}_{new}$, and fine-tuning adjusts a converged model $\mathcal{M}^{(\theta^*)}$ to fit a subset of its original task. Similarly, the computational cost of RELOAD, though similar to many baselines, is considerably greater than either SSD or gradient ascent. The results in Table 1 are produced using a ResNet-18 model on CIFAR-100; additional results with different models and datasets, and on randomly assigning 30% of training data samples to $\mathcal{D}_{forget}$ can be found in Appendix A.6.

| Method | NA ($\uparrow$) | $\Delta$FA ($\downarrow$) | $\Delta$FE ($\downarrow$) | $\Delta$FMIA ($\downarrow$) | Cost ($\downarrow$) | NSKL ($\downarrow$) | FSKL ($\downarrow$) |
|---|---|---|---|---|---|---|---|
| Retrain | $99.98_{\pm 0.01}$ | $74.89_{\pm 2.03}$ | $1.06_{\pm 0.13}$ | $0.63_{\pm 0.20}$ | $1.00_{\pm 0.00}$ | $0.00_{\pm 0.00}$ | $0.00_{\pm 0.00}$ |
| GA | $93.81_{\pm 0.75}$ | $18.77_{\pm 2.43}$ | $0.95_{\pm 0.14}$ | $0.21_{\pm 0.06}$ | $\mathbf{0.00_{\pm 0.00}}$ | $0.29_{\pm 0.09}$ | $2.62_{\pm 0.05}$ |
| FT | $96.00_{\pm 0.12}$ | $16.46_{\pm 2.47}$ | $0.89_{\pm 0.14}$ | $0.19_{\pm 0.08}$ | $0.27_{\pm 0.00}$ | $\mathbf{0.03_{\pm 0.01}}$ | $2.11_{\pm 0.06}$ |
| SSD | $1.01_{\pm 0.02}$ | $74.17_{\pm 2.04}$ | $4.19_{\pm 0.59}$ | $0.15_{\pm 0.21}$ | $0.01_{\pm 0.00}$ | $14.90_{\pm 1.24}$ | $11.81_{\pm 1.24}$ |
| SCRUB | $93.76_{\pm 0.74}$ | $18.85_{\pm 2.39}$ | $0.95_{\pm 0.14}$ | $0.20_{\pm 0.06}$ | $0.02_{\pm 0.00}$ | $0.29_{\pm 0.09}$ | $2.63_{\pm 0.06}$ |
| CF-$k$ | $94.75_{\pm 0.41}$ | $18.01_{\pm 2.60}$ | $0.94_{\pm 0.14}$ | $0.20_{\pm 0.06}$ | $0.21_{\pm 0.00}$ | $0.14_{\pm 0.03}$ | $2.47_{\pm 0.07}$ |
| EU-$k$ | $94.32_{\pm 0.49}$ | $17.93_{\pm 2.55}$ | $0.94_{\pm 0.14}$ | $0.20_{\pm 0.06}$ | $0.21_{\pm 0.00}$ | $0.19_{\pm 0.05}$ | $2.33_{\pm 0.05}$ |
| SalUn | $99.06_{\pm 0.22}$ | $13.14_{\pm 2.53}$ | $0.11_{\pm 0.09}$ | $7.39_{\pm 2.60}$ | $0.16_{\pm 0.00}$ | $0.06_{\pm 0.02}$ | $\mathbf{0.55_{\pm 0.04}}$ |
| Fisher | $97.76_{\pm 0.78}$ | $22.99_{\pm 2.30}$ | $0.95_{\pm 0.14}$ | $7.27_{\pm 2.48}$ | $1.78_{\pm 0.04}$ | $0.07_{\pm 0.02}$ | $0.56_{\pm 0.04}$ |
| RELOAD | $\mathbf{99.56_{\pm 0.11}}$ | $\mathbf{0.30_{\pm 0.50}}$ | $\mathbf{0.04_{\pm 0.02}}$ | $\mathbf{0.01_{\pm 0.01}}$ | $0.12_{\pm 0.01}$ | $0.15_{\pm 0.03}$ | $1.23_{\pm 0.11}$ |

Table 1: **10% Random Forgetting on CIFAR-100 (ResNet-18).** The top row presents the value of $\mathcal{M}^{(\theta^{\sim})}$ on each metric. Subsequent rows for $\Delta$FA ($\downarrow$), $\Delta$FE ($\downarrow$), and $\Delta$FMIA ($\downarrow$) present the absolute difference in the value of the corresponding method on this metric to the value of $\mathcal{M}^{(\theta^{\sim})}$ on the metric. These results show that RELOAD outperforms all the baselines on NA, $\Delta$FA, $\Delta$FE, $\Delta$FMIA, and FSKL by large margins. RELOAD performs competitively on the NSKL metric, outperformed by FT and CF-$k$. RELOAD incurs a higher computational cost than most baselines, but is cheaper than FT, CF-$k$, and EU-$k$.

| Method | NA ($\uparrow$) | FA ($\Delta^{\downarrow}$) | FE ($\Delta^{\downarrow}$) | FMIA ($\Delta^{\downarrow}$) | Cost ($\downarrow$) | NSKL ($\downarrow$) | FSKL ($\downarrow$) |
|---|---|---|---|---|---|---|---|
| Retrain | $99.99_{\pm 0.00}$ | $95.12_{\pm 0.23}$ | $0.20_{\pm 0.01}$ | $0.50_{\pm 0.00}$ | $1.00_{\pm 0.00}$ | $0.00_{\pm 0.00}$ | $0.00_{\pm 0.00}$ |
| GA | $99.57_{\pm 0.02}$ | $4.37_{\pm 0.25}$ | $0.17_{\pm 0.01}$ | $0.05_{\pm 0.01}$ | $\mathbf{0.00_{\pm 0.00}}$ | $0.05_{\pm 0.00}$ | $0.52_{\pm 0.02}$ |
| FT | $\mathbf{99.99_{\pm 0.00}}$ | $4.33_{\pm 0.22}$ | $0.17_{\pm 0.01}$ | $0.04_{\pm 0.01}$ | $0.27_{\pm 0.00}$ | $\mathbf{0.00_{\pm 0.00}}$ | $0.43_{\pm 0.02}$ |
| SSD | $12.75_{\pm 4.69}$ | $82.52_{\pm 4.73}$ | $2.12_{\pm 0.06}$ | $0.01_{\pm 0.01}$ | $0.01_{\pm 0.00}$ | $8.55_{\pm 0.13}$ | $7.88_{\pm 0.12}$ |
| SCRUB | $99.79_{\pm 0.01}$ | $4.44_{\pm 0.26}$ | $0.18_{\pm 0.01}$ | $0.05_{\pm 0.01}$ | $0.03_{\pm 0.00}$ | $0.03_{\pm 0.00}$ | $0.50_{\pm 0.02}$ |
| CF-$k$ | $99.76_{\pm 0.01}$ | $4.47_{\pm 0.24}$ | $0.18_{\pm 0.01}$ | $0.05_{\pm 0.01}$ | $0.23_{\pm 0.02}$ | $0.03_{\pm 0.00}$ | $0.50_{\pm 0.02}$ |
| EU-$k$ | $99.63_{\pm 0.01}$ | $4.46_{\pm 0.25}$ | $0.18_{\pm 0.01}$ | $0.05_{\pm 0.01}$ | $0.23_{\pm 0.02}$ | $0.05_{\pm 0.00}$ | $0.47_{\pm 0.02}$ |
| SalUn | $99.90_{\pm 0.04}$ | $3.14_{\pm 1.00}$ | $0.13_{\pm 0.03}$ | $0.04_{\pm 0.02}$ | $0.17_{\pm 0.00}$ | $0.03_{\pm 0.00}$ | $0.50_{\pm 0.02}$ |
| Fisher | $99.57_{\pm 0.02}$ | $\mathbf{0.09_{\pm 0.05}}$ | $\mathbf{0.00_{\pm 0.00}}$ | $0.01_{\pm 0.00}$ | $2.17_{\pm 0.04}$ | $0.05_{\pm 0.00}$ | $0.47_{\pm 0.02}$ |
| RELOAD | $99.68_{\pm 0.17}$ | $0.25_{\pm 0.21}$ | $0.01_{\pm 0.01}$ | $\mathbf{0.00_{\pm 0.00}}$ | $0.12_{\pm 0.01}$ | $0.06_{\pm 0.02}$ | $\mathbf{0.21_{\pm 0.02}}$ |

Table 2: **100 In Class Random Forgetting on SVHN (ResNet-18)**
$\uparrow$: the goal is to have as high of a value as possible, $\Delta^{\downarrow}$: the value in the table is the difference between the result of the unlearning method and retraining (top row) on the metric and the goal is to have a low difference, $\downarrow$: the goal is to have as low of a value as possible. The top row presents the value of $\mathcal{M}^{(\theta^{\sim})}$ on each metric. Subsequent rows for $\Delta$FA ($\downarrow$), $\Delta$FE ($\downarrow$), and $\Delta$FMIA ($\downarrow$) present the absolute difference in the value of the corresponding method on this metric to the value of $\mathcal{M}^{(\theta^{\sim})}$ on the metric. These results show that RELOAD outperforms all the baselines on $\Delta$FA, $\Delta$FE, $\Delta$FMIA, and NSKL by large margins. RELOAD performs competitively on NA and FSKL but is outperformed by FT. RELOAD also incurs a higher computational cost than the other baselines.

**RELOAD efficiently unlearns correlated samples.** We next randomly assign 100 samples from a single class of the training data to $\mathcal{D}_{forget}$, to evaluate how well each method can unlearn arbitrary but related training samples. The results of this experiment are shown in Table 2. RELOAD achieves the lowest $\Delta$FMIA and FSKL of all approaches and very close to the lowest $\Delta$FA, $\Delta$FE, and NSKL of all approaches, suggesting that again, RELOAD learns to closely approximate $\mathcal{M}^{(\theta^{\sim})}$ in this setting. RELOAD is marginally outperformed

Original Image Image with Backdoor

Figure 3: The "Cross Pattern Backdoor" inserts the above pattern (*right*) in all images from

by Fisher in these settings, but is far more realistic, as Fisher over twice as much time as retraining. Although RELOAD achieves an NA competitive with that of most baselines, naive gradient ascent, CF-$k$, and EU-$k$ yield a marginally higher NA; this is surprising for gradient ascent as it typically yields lower NA values. This can be attributed to the small number of unlearning samples; optimizing to maximize the loss on these samples does not provide much of a gradient update. CF-$k$ and EU-$k$ both make few parameter updates to $\mathcal{M}^{(\theta^*)}$, which leads to a high NA but poor performance on unlearning metrics like $\Delta$FA and $\Delta$FE.

## 4.3 REMEDIAL LEARNING EXPERIMENTS

**Baselines.** The remedial unlearning setting admits different baselines than the unlearning setting. "Original" represents a baseline model trained on $\mathcal{D}$, and "Retrain" represents a baseline model trained directly on $\mathcal{D}_{new}$. Then, because gradient ascent does not directly translate to the task of remedial learning (because there is no "forget set" on which to ascent) we introduce two variants of gradient ascent to serve as baselines. Gradient Ascent Relearn (GAR) performs 10 epochs of gradient ascent on $\mathcal{D}$, followed by 10 epochs of gradient descent on $\mathcal{D}_{new}$. Gradient Difference Ascent (GRDA) calculates $\nabla \mathcal{L}(\mathcal{D}) - \nabla \mathcal{L}(\mathcal{D}_{new})$ on each step and performs a gradient update in the opposite direction, fitting $\mathcal{D}_{new}$. It performs 10 epochs of such updates.

**Metrics.** We evaluate remedial learning as follows. The accuracy on a held-out test split of $\mathcal{D}_{new}$ (Acc. $\mathcal{D}_{new}^{(test)}$, $\uparrow$) represents how well the model fits the distribution of $\mathcal{D}_{new}$ (without the backdoor; see next section). The accuracy on a held-out test spit of $\mathcal{D}_{new}$ with backdoors added to each instance (Acc. $\mathcal{D}_{new}^{(test,\S)}$) measures the reliance of the model on the backdoor pattern in classification. Specifically, a low accuracy on $\mathcal{D}_{new}^{(test,\S)}$ suggests that the model is over-reliant on the backdoor pattern that was injected into its training data.

**RELOAD corrects erroneous data (removing shortcuts).** In this setting, we select 2 classes from the data (here, CIFAR-10) and inject cross-patterns into the corners of their training samples to construct $\mathcal{D}$. An example of this backdoor can be seen in Figure 3. The inclusion of this backdoor influences a model trained on this dataset to rely on the cross-patterns as strong indicators of class membership. To construct $\mathcal{D}_{new}$ we then replace the cross-patterned samples with their original instances, removing the backdoor. The goal of remedial learning here is to un-learn the model's reliance on the backdoor, and re-learn the salient representations needed to accurately predict on the affected classes. The results of this experiment are shown in Appendix A.9. Observe that the effect of this backdoor attack produces a trained model (Original; trained *with* the backdoor) with poor performance on $\mathcal{D}_{new}^{(test,\S)}$,

| Method | Acc. $\mathcal{D}_{new}^{(test)}$ ($\uparrow$) | Acc. $\mathcal{D}_{new}^{(test,\S)}$ ($\uparrow$) | Cost ($\downarrow$) |
|---|---|---|---|
| Original | $82.68_{\pm 0.45}$ | $19.81_{\pm 0.03}$ | N/A |
| Retrain | $92.48_{\pm 0.00}$ | $91.90_{\pm 0.00}$ | $1.00_{\pm 0.00}$ |
| GAR | $57.29_{\pm 34.88}$ | $56.54_{\pm 34.12}$ | $0.08_{\pm 0.01}$ |
| GRDA | $62.87_{\pm 28.47}$ | $62.34_{\pm 27.80}$ | $0.05_{\pm 0.00}$ |
| FT | $86.87_{\pm 4.39}$ | $86.50_{\pm 4.14}$ | $0.37_{\pm 0.02}$ |
| SSD | $30.25_{\pm 22.90}$ | $23.94_{\pm 13.43}$ | $\mathbf{0.01_{\pm 0.00}}$ |
| SCRUB | $12.43_{\pm 3.45}$ | $12.42_{\pm 3.44}$ | $0.04_{\pm 0.01}$ |
| CF-$k$ | $66.56_{\pm 24.27}$ | $66.29_{\pm 23.80}$ | $0.29_{\pm 0.03}$ |
| EU-$k$ | $66.75_{\pm 24.08}$ | $66.41_{\pm 23.63}$ | $0.29_{\pm 0.03}$ |
| RELOAD | $\mathbf{90.81_{\pm 0.99}}$ | $\mathbf{90.51_{\pm 0.82}}$ | $0.08_{\pm 0.06}$ |

Table 3: **Cross Pattern Backdoor Removal on CIFAR-10 (ResNet-18).** $\uparrow$: the goal is to have as high of a value as possible, $\downarrow$: the goal is to have as low of a value as possible. These results show that RELOAD outperforms all baselines on Acc. $\mathcal{D}_{new}^{(test)}$ and Acc. $\mathcal{D}_{new}^{(test,\S)}$. The small differences between these accuracy values for RELOAD indicate that it successfully removed the influence of the backdoor pattern. RELOAD incurs a higher computational cost than most baselines, but is cheaper than FT, CF-$k$, and EU-$k$.

because the model learned to treat the backdoor pattern as a strong indicator of class membership for certain classes. Further, notice that RELOAD successfully remedies this vulnerability, achieving the highest accuracy (aside from Retrain; retrained from scratch *without* the backdoor) on $\mathcal{D}_{new}^{(test)}$, and the highest accuracy of all models on $\mathcal{D}_{new}^{(test,\S)}$. This suggests that RELOAD is capable of efficiently correcting the predictive behaviour of a model trained on erroneous data.

## 5 DISCUSSION

This work introduces the setting of *blind unlearning*, machine unlearning without direct access to the "forget set". This setting allows for improved privacy procedures in practical settings, by enabling the immediate deletion of data when an unlearning request is received rather than retaining the data for the purpose of downstream unlearning. Our method, RELOAD, combines insights from gradient-based unlearning (to remove top-level information from all parameters) with selective parameter re-initialization. The blind setting ensures that as long as practitioners store the last step gradients of their model on the training set, they have the capacity to unlearn data when it is removed from their system. We recommend that future work study the performance of RELOAD at larger scales, such as those presented by modern large language models (Achiam et al., 2023), and investigate the utility of other choices for $\mathcal{I}_\mathcal{D}$ beyond the cached gradients used in RELOAD.

Despite operating in the blind setting, RELOAD outperforms benchmark machine unlearning algorithms that enjoy direct access to $\mathcal{D}_{forget}$, suggesting that it is an empirically effective unlearning algorithm. However, RELOAD admits a modest tradeoff between computational efficiency and performance in this regime. We finally observe that machine unlearning represents a special case of *remedial learning*, a setting that is especially important for efficiently correcting errors in the training data used to train models. RELOAD remains an efficient, performant method in this regime, suggesting that our work may contain generalizable insights about about learning to fit arbitrary downstream transformations of data.

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
