# A  APPENDIX

## A.1  REMEDIAL LEARNING GRADIENT DERIVATION

In the setting of Remedial Learning, we construct these sets.

$$\mathcal{D} \cap \mathcal{D}_{new}^c = \{(X_i, Y_i)\}_{i=K+1\ldots N}, \mathcal{D}^c \cap \mathcal{D}_{new} = \{(X'_i, Y'_i)\}_{i=K+1\ldots N}, \text{ and} \tag{9}$$

$$\mathcal{D} \cap \mathcal{D}_{new} = \{(X_i, Y_i)\}_{i=1\ldots K} \tag{10}$$

The gradient then formulates as

$$\sum_{\substack{(X_i,Y_i)\in \\ \{(X_i,Y_i)\}_{i=K+1\ldots N}}} \nabla_\theta \mathcal{L}((X_i, Y_i), \hat{Y}_i) - \sum_{\substack{(X_i,Y_i)\in \\ \{(X'_i,Y'_i)\}_{i=K+1\ldots N}}} \nabla_\theta \mathcal{L}((X'_i, Y'_i), \hat{Y}_i) = \sum_{(X_i,Y_i)\in\mathcal{D}} \nabla_\theta \mathcal{L}((X_i, Y_i), \hat{Y}_i) \tag{11}$$

$$- \mathbb{1}_{\substack{(X_i,Y_i)\in \\ \mathcal{D}\cap\mathcal{D}_{new}}} \Big[ \nabla_\theta \mathcal{L}((X_i, Y_i), \hat{Y}_i) \Big] - \Big( \sum_{(X_i,Y_i)\in\mathcal{D}_{new}} \nabla_\theta \mathcal{L}((X_i, Y_i), \hat{Y}_i) - \mathbb{1}_{\substack{(X_i,Y_i)\in \\ \mathcal{D}\cap\mathcal{D}_{new}}} \Big[ \nabla_\theta \mathcal{L}((X_i, Y_i), \hat{Y}_i) \Big] \Big) \tag{12}$$

$$= \sum_{(X_i,Y_i)\in\mathcal{D}} \nabla_\theta \mathcal{L}((X_i, Y_i), \hat{Y}_i) - \sum_{(X_i,Y_i)\in\mathcal{D}\cap\mathcal{D}_{new}} \nabla_\theta \mathcal{L}((X_i, Y_i), \hat{Y}_i) - \sum_{(X_i,Y_i)\in\mathcal{D}_{new}} \nabla_\theta \mathcal{L}((X_i, Y_i), \hat{Y}_i) + \sum_{(X_i,Y_i)\in\mathcal{D}\cap\mathcal{D}_{new}} \nabla_\theta \mathcal{L}((X_i, Y_i), \hat{Y}_i) \tag{13}$$

$$= \sum_{(X_i,Y_i)\in\mathcal{D}} \nabla_\theta \mathcal{L}((X_i, Y_i), \hat{Y}_i) - \sum_{(X_i,Y_i)\in\mathcal{D}_{new}} \nabla_\theta \mathcal{L}((X_i, Y_i), \hat{Y}_i) \tag{14}$$

$$= \nabla_\theta \mathcal{L}(\mathcal{D}) - \nabla_\theta \mathcal{L}(\mathcal{D}_{new}) \tag{15}$$

## A.2  FURTHER ARGUMENT FOR NON-RECOVERABILITY FROM GRADIENTS

We discuss the behaviour for non-softmax activated models. In this case, a single output logit of the model can be inferred in the $|\mathcal{D}_{forget}| = 1$ case. If $|\mathcal{D}_{forget}| > 1$, the summation operation makes the data non-recoverable. In this case, the $j$-th component map $\mathcal{M}^{(\theta^*)}(x)$ is a non-linear function $F : \mathbb{R}^d \to \mathbb{R}$. In this case the non-recoverability of the gradient depends on the model, and its invertability, specifically the component map $F$. If $F$ is bijective, then this violates our definition of non-recoverability, and if $F$ is injective, then the input data $x$ can be recovered by performing a search, as only a single data point will produce this gradient. However, classification models without the softmax function are rare, and within that subset, those which are invertible or injective are even rarer. Additionally, within the small subset of models for which these conditions apply, a large, exhaustive, and computationally intensive search over the input space will still be required (in the injective but not surjective case) to recover the input $x$.

Additionally, we can consider other models trained with different loss functions. In these landscapes the recoverability of the model depends first on the injectivity of the derivative of the loss function with respect to $\mathcal{D}_{forget}$ when $|\mathcal{D}_{forget}| = 1$ amd when $|\mathcal{D}_{forget}| > 1$. For example, for models trained with MSE, the derivative of the loss function is $\frac{2}{N} \sum_{x_i \in \mathcal{D}_{forget}} (y_i - \mathcal{M}^{(\theta^*)}(x_i))$, which is a summation over differences - inherently not injective in both cases and thus does not permit recoverability of data.

## A.3  EXPERIMENTAL DETAILS

In this section we detail our experimental setup for the RELOAD algorithm. We carry out a set of empirical evaluations of the method, comparing it against other state-of-the-art unlearning baselines. The empirical metrics we consider for unlearning are detailed in Table 6, and the metrics for remedial learning are detailed in Table 7.

We train ResNet-18 and VGG16-BN models on CIFAR-10 (Krizhevsky, 2012), CIFAR-100 (Krizhevsky et al.), and SVHN (Netzer et al., 2011) for image classification for 182 epochs. We apply the cross-entropy loss function and a learning rate of 0.1 with a batch size of 256. We conducted these experiments over 10 random seeds to obtain average results and standard deviation measurements. The results in our tables are reported in the format $\mu_{\pm\sigma}$ where $\mu$ is the average value and $\sigma$ is the standard deviation, across the 10 seeds.

The 10 seeds we selected for unlearning experiments were seeds $\{1, 2, 3, 4, 5, 6, 7, 8, 9, 10\}$ and for remedial learning we used seeds $\{0, 1, 2, 3, 4, 5, 6, 7, 8, 9\}$.

**Hardware.** All experiments were run on 4 CPU cores, 20 GB of RAM, and 1 NVIDIA T4 GPU.

**Baseline Implementations.** Implementations for baselines were taken from the reference implementations for SCRUB, SSD, EU-$k$, and CF-$k$. Implementations for FT and GA were taken from the repository for SalUn.

**Hyperparameters.** Hyperparameters for RELOAD were chosen through a hyperparameter sweep. The chosen hyperparameters for the unlearning tasks are presented in Table 4 and the hyperparameters for the remedial learning tasks are presented in Table 5.

We empirically find that the cumulative distribution function of the knowledge-values for forgetting 10% of data from a ResNet-18 model trained on SVHN forms a sigmoid-like curve around $10^{-1}$. This further evidences the existence of clear differences in the knowledge-values for different parameters. Experimentally, we select the thresholding hyperparameter $\alpha$ using a hyperparameter sweep. We have included in ablation (Appendix **??**), a study with varying learning rates ($\eta$) and thresholds ($\alpha$).

| Experiment | Alpha ($\alpha$) | Priming Learning Rate | Retraining Learning Rate |
|---|---|---|---|
| SVHN + ResNet-18 | 0.1 | 0.243 | 0.098 |
| SVHN + VGG16-BN | 0.1 | 0.496 | 0.496 |
| CIFAR-10 + ResNet-18 | 0.1 | 0.44 | 0.33 |
| CIFAR-10 + VGG16-BN | 0.1 | 0.167 | 0.39 |
| CIFAR-100 + ResNet-18 | 0.1 | 0.18 | 0.33 |
| CIFAR-100 + VGG16-BN | 0.1 | 0.325 | 0.164 |

Table 4: Hyperparameter Settings for Unlearning

| Experiment | Alpha ($\alpha$) | Priming Learning Rate | Retraining Learning Rate |
|---|---|---|---|
| SVHN + ResNet-18 | 0.13 | 0.068 | 0.365 |
| SVHN + VGG16-BN | 0.14 | 0.14 | 0.195 |
| CIFAR-10 + ResNet-18 | 0.147 | 0.074 | 0.415 |
| CIFAR-10 + VGG16-BN | 0.27 | 0.106 | 0.278 |
| CIFAR-100 + ResNet-18 | 0.16 | 0.09 | 0.136 |
| CIFAR-100 + VGG16-BN | 0.22 | 0.173 | 0.103 |

Table 5: Hyperparameter Settings for Remedial Learning

**Unlearning Evaluation Metrics**

| Statistic | Abbr. | Description |
|---|---|---|
| Accuracy on $dataset_{new}$ ($\uparrow$) | NA | Model accuracy on the $\mathcal{D}_{new}$. In unlearning, a higher accuracy indicates that the unlearning process has not negatively impacted the model's performance on the retained data. |
| Diff. in Accuracy on $\mathcal{D}_{forget}$ ($\downarrow$) | $\Delta$FA | The change in accuracy on the forget set between the current model and $\mathcal{M}^{(\theta^\sim)}$. A smaller difference, approaching the accuracy of the retrained model, indicates that the unlearning method has been more effective in "forgetting" the forget set. |
| Diff. in Error on $\mathcal{D}_{forget}$ ($\downarrow$) | $\Delta$FE | The reduction in error on the forget set between the current model and $\mathcal{M}^{(\theta^\sim)}$. A smaller difference, approaching the error of the retrained model, signifies that the unlearning method has been more effective at "forgetting" the forget set. |
| Diff. in MIA Success Rate on $\mathcal{D}_{forget}$ ($\downarrow$) | $\Delta$FMIA | Difference in success rate of a membership inference attack (MIA) on the forget set between the current model and $\mathcal{M}^{(\theta^\sim)}$. In this work, we use the attack from Shokri et al. (2017) implemented in the repository for Kurmanji et al. (2023). A success rate approaching that of the retrained model implies the forgotten data is indistinguishable to an MIA on in-distribution data that the model was not trained on. |
| Symmetric KL-Divergence on $\mathcal{D}_{new}$ ($\downarrow$) | NSKL | Symmetric KL-Divergence between the logits of the current model and those of $\mathcal{M}^{(\theta^\sim)}$. This metric is averaged over all instances in the $\mathcal{D}_{new}$. A lower Symmetric KL divergence indicates an unlearning method that behaves similarly on the $\mathcal{D}_{new}$ to a model retrained from scratch without the forget set. |
| Symmetric KL-Divergence on $\mathcal{D}_{forget}$ ($\downarrow$) | FSKL | The Symmetric KL-Divergence between the logits of the current model and those of $\mathcal{M}^{(\theta^\sim)}$. This metric is averaged over all instances in the $\mathcal{D}_{forget}$. A lower Symmetric KL divergence indicates that the unlearning method that behaves similarly on the $\mathcal{D}_{forget}$ to a model retrained from scratch without the forget set. |
| Cost ($\downarrow$) | Cost | Ratio of the runtime of the unlearning method to the runtime of retraining a baseline model from scratch without the forget set. A lower cost indicates a more computationally efficient method. |

Table 6: **Evaluation Statistics for Unlearning.**

**Remedial Learning Evaluation Metrics**

| Statistic | Abbr. | Description |
|---|---|---|
| Accuracy on $\mathcal{D}_{new}$ ($\uparrow$) | NA | Model accuracy on $\mathcal{D}_{new}$. In remedial learning, a higher accuracy indicates that the remedial learning process has correctly adapted the model to its new training set. |
| Accuracy on $\mathcal{D}$ ($\uparrow$) | OA | Model accuracy on $\mathcal{D}$. In the case of backdoor attacks or noisy remedial learning, a higher value indicates the relearned model correctly has lost its reliance on the backdoor pattern. In label correction setting, the desirable value is the percentage of samples that did not have their labels flipped (in our experiments, 90%). |
| Accuracy on $\mathcal{D}_{new}^{(test)}$ ($\uparrow$) | TA | Model accuracy on a held out test-set. A higher accuracy indicates that the relearned model generalizes well to in-distribution tasks outside of its old and new training set. |
| Accuracy on Transformed $\mathcal{D}_{new}^{(\S)}$ ($\uparrow$) | TNA | Model accuracy on $\mathcal{D}_{new}$ with backdoors added to each instance. A higher accuracy indicates that the relearned model does not rely on the presence of the backdoor to make its inference, and that despite the presence of the backdoor, it correctly classifies. |
| Accuracy on $\mathcal{D}_{new}^{(test,\S)}$ ($\uparrow$) | TTA | Model accuracy on $\mathcal{D}_{new}^{(test)}$ with backdoors added to each instance. A higher accuracy indicates that the relearned model does not rely on the presence of the backdoor to make its inference, and that despite the presence of the backdoor, it correctly classifies data that is in-distribution but outside of its old and new training sets. |
| Cost ($\downarrow$) | Cost | Ratio of the runtime of the remedial learning method to the runtime of retraining a baseline model from scratch without the forget set. A lower cost indicates a more computationally efficient method. |

Table 7: **Evaluation Statistics for Remedial Learning.**

**Mislabelling Class Pairings**

In the below tables we list out the semantically similar classes we chose for each of the 3 datasets, CIFAR-10, CIFAR-100, and SVHN, to perform targeted mislabelling attacks against.

| Original Class | Original Label | Flipped Class | Flipped Label |
|:---:|:---:|:---:|:---:|
| 0 | airplane | 2 | bird |
| 2 | bird | 0 | airplane |
| 3 | cat | 5 | dog |
| 5 | dog | 3 | cat |
| 1 | automobile | 9 | truck |
| 9 | truck | 1 | automobile |

Table 8: Flip mappings for CIFAR-10 with class labels

| Superclass | Original Class | Original Label | Flipped Class | Flipped Label |
|:---:|:---:|:---:|:---:|:---:|
| Aquatic mammals | 0 | beaver | 1 | dolphin |
| Flowers | 50 | orchid | 53 | sunflower |
| Insects | 75 | bee | 77 | butterfly |
| Vehicles 1 | 60 | bicycle | 61 | bus |
| Large carnivores | 17 | lion | 18 | tiger |
| Large omnivores/herbivores | 24 | cattle | 26 | elephant |
| Small mammals | 37 | mouse | 38 | rabbit |
| Fruits and vegetables | 82 | apple | 83 | mushroom |
| Household furniture | 56 | chair | 58 | table |
| Trees | 92 | maple tree | 93 | oak tree |

Table 9: Flip mappings for CIFAR-100 with class labels

| Original Class | Flipped Class |
|:---:|:---:|
| 0 | 6 |
| 1 | 7 |
| 2 | 5 |
| 3 | 8 |
| 4 | 9 |
| 5 | 3 |
| 6 | 0 |
| 7 | 1 |
| 8 | 3 |
| 9 | 4 |

Table 10: Flip mappings for SVHN

## A.4 REMEDIAL LEARNING JUSTIFICATION

Machine learning models mimic their training data, and as such data which is incorrectly labelled, contains transformed samples (eg. backdoor-injected samples), is biased, or is corrupted can make a huge impact on the downstream performance of a model.

Aside from existing label-flip attacks, backdoor attacks, and the possibility of corrupted data, there is also the need to account for human error. To apply supervised machine learning algorithms large amounts of data need to be properly labeled for the learning procedure. This is not always feasible under a budget, and human labeling is not error-free. Ho-Phuoc (2018) shows that human annotation on CIFAR-10 has an accuracy of $94.91\%$. Crowdsourcing labels is also not a reliable approach due to human errors, or potentially adversarial attacks through mislabelling (Lin et al., 2021). In remedial learning, we assume that a subset of the labelled training data is incorrect, and that a labeler

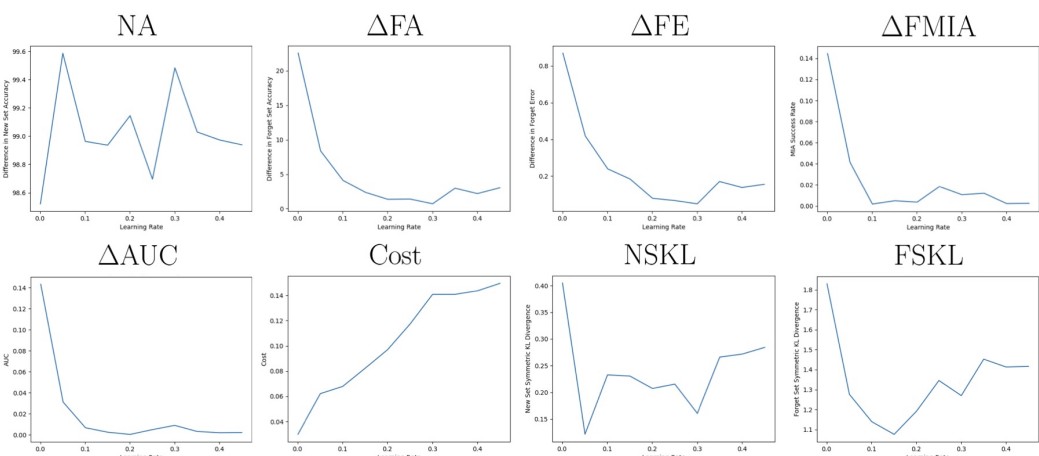

Figure 4: Impact of Learning Rate ($\eta$) on RELOAD performance

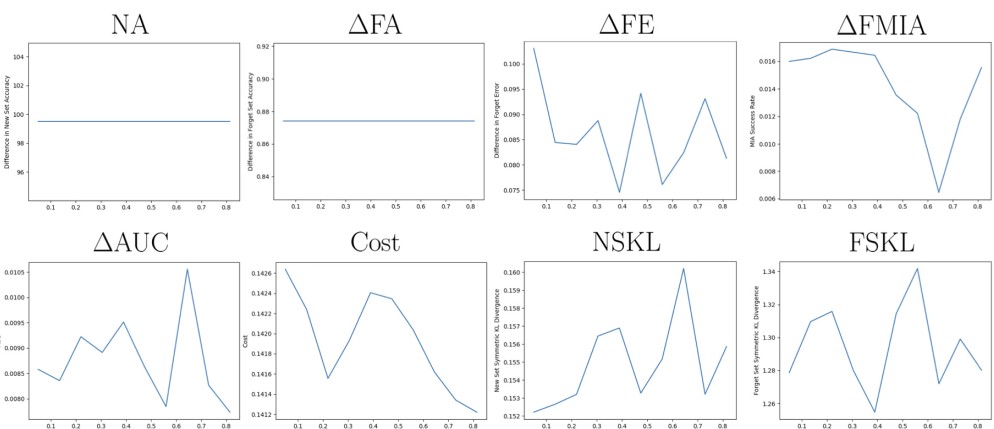

Figure 5: Impact of Threshold ($\alpha$) on RELOAD performance

mislabels data a percentage of the time. As demonstrated by Fard et al. (2017), biased labelling can greatly damage the classification accuracy of a target class with little effect on the other classes. This provides the motivation for studying the case of remedial learning.

## A.5 ABLATION STUDIES

### A.5.1 LEARNING RATE $\eta$ AND THRESHOLD $\alpha$

We study the effect of different learning rates on the unlearning performance exhibited by the RELOAD algorithm. For this study, we select the case of randomly forgetting 10% of the training data from a ResNet-18 model trained on CIFAR-100.

As shown in Figure 4, we observe that the choice of learning rate has a significant impact on performance. This is particularly true in the case of $\Delta$FA, $\Delta$FE, $\Delta$FMIA, and $\Delta$AUC measurements - which are the primary metrics evaluating how well the model has forgotten $\mathcal{D}_{forget}$. Based on these plots, we choose $\eta = 0.33$.

Figure 5 shows the effect of varying the proportion of the parameters that are selected for reinitialisation ($\alpha$). We observe that the choice of threshold has an impact on the performance of the RELOAD

algorithm and that its selection involves a tradeoff between the different metrics we consider. Thus, the best choice of $\alpha$ should ideally be selected through a hyperparameter search.

### A.5.2 METHODS OF SELECTING KNOWLEDGEABLE PARAMETERS

In designing the knowledge values for identifying knowledgeable parameters, we considered several other approaches in addition to the final formula 8. This includes the salient weight formula first introduced by Fan et al. (2023), in choosing the parameters with the highest magnitude gradients on $\mathcal{D} - \mathcal{D}_{new}$. Empirically, forgetting was not properly achieved, and in some cases re-initializing these parameters caused model collapse.

Secondly, as performed by Foster et al. (2023), we considered the importance value produced by an approximation to the Fisher Information Matrix. This increased computational overhead and produced similar but slightly poorer on average results. Thirdly, we took inspiration from Hassibi et al. (1993) and designed a 2nd-order hessian-based formula (Equation 16) which we only studied in the unlearning case. This approach offered no noticeable performance increase and drastically increased computational overhead on Hessian computing, even with KFAC (Martens & Grosse, 2015) and EKFAC (Gao et al., 2020) approximations. This made this method infeasible.

$$\frac{1}{2}\delta W^T (H_{\mathcal{D}_{new}} - H_{\mathcal{D}_{forget}}) \cdot \delta W \tag{16}$$

### A.5.3 PRIMING STEPS

In designing the priming step, we considered the possibility of needing multiple steps to appropriately scrub the global information from the model parameters. Theoretically, this notion violates the blind nature of the unlearning setup, and was thus undesirable. Empirically, we noted that using multiple priming steps does not improve forgetting and can lead to further performance degradation on $\mathcal{D}_{new}$ requiring more retraining to get to a final unlearned model.

### A.5.4 RETRAINING METHODS

Aside from using the classic training setup that was originally used to train the model, we considered a teacher-student setup to speed up retraining. We use the original model $\mathcal{M}^{(\theta^*)}$ as the teacher and the re-initialised model as the student. In theory distillation training is faster. Empirically, this process reached the same target downstream performance as classic training in the same amount of time, and yielded poorer forgetting results measured on $\Delta$FA. We hypothesize that through distilling on $\mathcal{D}_{new}$, implicit knowledge about $\mathcal{D}_{forget}$ in $\mathcal{M}^{(\theta^*)}$, is taught and recovered by the re-initialised model.

## A.6 RANDOM 10% FORGETTING - ADDITIONAL EXPERIMENTS

| Method | NA ($\uparrow$) | $\Delta$FA ($\downarrow$) | $\Delta$FE ($\downarrow$) | $\Delta$FMIA ($\downarrow$) | Cost ($\downarrow$) | NSKL ($\downarrow$) | FSKL ($\downarrow$) |
|---|---|---|---|---|---|---|---|
| Retrain | $99.60_{\pm 0.07}$ | $91.88_{\pm 0.70}$ | $0.08_{\pm 0.12}$ | $0.54_{\pm 0.02}$ | $1.00_{\pm 0.00}$ | $0.00_{\pm 0.00}$ | $0.00_{\pm 0.00}$ |
| GA | $98.41_{\pm 0.24}$ | $7.15_{\pm 0.72}$ | $0.06_{\pm 0.13}$ | $\mathbf{0.01_{\pm 0.02}}$ | $\mathbf{0.00_{\pm 0.00}}$ | $0.06_{\pm 0.03}$ | $0.67_{\pm 0.06}$ |
| FT | $98.24_{\pm 0.19}$ | $3.83_{\pm 0.41}$ | $0.12_{\pm 0.01}$ | $0.02_{\pm 0.00}$ | $0.27_{\pm 0.01}$ | $\mathbf{0.05_{\pm 0.01}}$ | $\mathbf{0.49_{\pm 0.04}}$ |
| SSD | $22.87_{\pm 34.01}$ | $70.85_{\pm 29.15}$ | $2.08_{\pm 0.83}$ | $0.04_{\pm 0.01}$ | $\mathbf{0.00_{\pm 0.00}}$ | $7.99_{\pm 3.52}$ | $7.56_{\pm 3.06}$ |
| SCRUB | $98.43_{\pm 0.23}$ | $7.17_{\pm 0.63}$ | $\mathbf{0.05_{\pm 0.13}}$ | $\mathbf{0.01_{\pm 0.02}}$ | $0.02_{\pm 0.00}$ | $0.06_{\pm 0.02}$ | $0.66_{\pm 0.04}$ |
| CF-$k$ | $98.31_{\pm 0.27}$ | $7.22_{\pm 0.69}$ | $0.06_{\pm 0.13}$ | $\mathbf{0.01_{\pm 0.02}}$ | $0.29_{\pm 0.01}$ | $0.06_{\pm 0.02}$ | $0.56_{\pm 0.04}$ |
| EU-$k$ | $98.35_{\pm 0.25}$ | $7.22_{\pm 0.71}$ | $0.06_{\pm 0.13}$ | $\mathbf{0.01_{\pm 0.02}}$ | $0.29_{\pm 0.01}$ | $0.06_{\pm 0.02}$ | $0.57_{\pm 0.04}$ |
| SalUn | $99.83_{\pm 0.05}$ | $0.33_{\pm 0.18}$ | $0.10_{\pm 0.01}$ | $0.01_{\pm 0.00}$ | $0.14_{\pm 0.00}$ | $0.06_{\pm 0.02}$ | $0.56_{\pm 0.04}$ |
| Fisher | $99.40_{\pm 0.22}$ | $4.28_{\pm 0.40}$ | $0.12_{\pm 0.01}$ | $0.02_{\pm 0.00}$ | $1.08_{\pm 0.03}$ | $0.06_{\pm 0.02}$ | $0.57_{\pm 0.03}$ |
| RELOAD | $\mathbf{99.48_{\pm 0.11}}$ | $\mathbf{2.20_{\pm 0.58}}$ | $0.30_{\pm 0.13}$ | $0.04_{\pm 0.01}$ | $0.34_{\pm 0.11}$ | $0.12_{\pm 0.01}$ | $0.54_{\pm 0.08}$ |

Table 11: **10% Random Forgetting on CIFAR-10 (VGG16-BN)**
$\uparrow$: the goal is to have as high of a value as possible, $\Delta^{\downarrow}$: the value in the table is the difference between the result of the unlearning method and retraining (top row) on the metric and the goal is to have a low difference, $\downarrow$: the goal is to have as low of a value as possible. The top row presents the value of $\mathcal{M}^{(\theta^{\sim})}$ on each metric. Subsequent rows for $\Delta$FA ($\downarrow$), $\Delta$FE ($\downarrow$), and $\Delta$FMIA ($\downarrow$) present the absolute difference in the value of the corresponding method on this metric to the value of $\mathcal{M}^{(\theta^{\sim})}$ on the metric. These results show that RELOAD outperforms all the baselines on NA, $\Delta$FA by large margins. RELOAD performs competitively on the $\Delta$FE, $\Delta$FMIA, FSKL, and NSKL metrics, but is outperformed. RELOAD incurs a higher computational cost than other baselines, but performs better across all metrics than other baselines.

| Method | NA ($\uparrow$) | $\Delta$FA ($\downarrow$) | $\Delta$FE ($\downarrow$) | $\Delta$FMIA ($\downarrow$) | Cost ($\downarrow$) | NSKL ($\downarrow$) | FSKL ($\downarrow$) |
|---|---|---|---|---|---|---|---|
| Retrain | $99.99_{\pm 0.01}$ | $94.40_{\pm 0.72}$ | $0.23_{\pm 0.08}$ | $0.50_{\pm 0.01}$ | $1.00_{\pm 0.00}$ | $0.00_{\pm 0.00}$ | $0.00_{\pm 0.00}$ |
| GA | $98.38_{\pm 0.21}$ | $3.86_{\pm 0.66}$ | $0.21_{\pm 0.07}$ | $0.04_{\pm 0.02}$ | $\mathbf{0.00_{\pm 0.00}}$ | $0.06_{\pm 0.02}$ | $0.66_{\pm 0.06}$ |
| FT | $98.24_{\pm 0.21}$ | $\mathbf{1.45_{\pm 0.53}}$ | $0.16_{\pm 0.03}$ | $0.03_{\pm 0.01}$ | $0.27_{\pm 0.00}$ | $\mathbf{0.05_{\pm 0.01}}$ | $\mathbf{0.48_{\pm 0.04}}$ |
| SSD | $20.02_{\pm 29.99}$ | $75.65_{\pm 26.45}$ | $1.88_{\pm 0.62}$ | $0.01_{\pm 0.02}$ | $0.01_{\pm 0.00}$ | $8.30_{\pm 3.11}$ | $7.83_{\pm 2.70}$ |
| SCRUB | $98.41_{\pm 0.20}$ | $3.89_{\pm 0.70}$ | $0.21_{\pm 0.07}$ | $0.04_{\pm 0.02}$ | $0.02_{\pm 0.00}$ | $0.06_{\pm 0.02}$ | $0.65_{\pm 0.04}$ |
| CF-$k$ | $98.28_{\pm 0.23}$ | $3.81_{\pm 0.71}$ | $0.21_{\pm 0.07}$ | $0.05_{\pm 0.02}$ | $0.21_{\pm 0.00}$ | $0.06_{\pm 0.02}$ | $0.55_{\pm 0.04}$ |
| EU-$k$ | $98.31_{\pm 0.21}$ | $3.83_{\pm 0.71}$ | $0.21_{\pm 0.07}$ | $0.05_{\pm 0.02}$ | $0.21_{\pm 0.00}$ | $0.07_{\pm 0.02}$ | $0.56_{\pm 0.04}$ |
| SalUn | $99.78_{\pm 0.05}$ | $3.68_{\pm 0.48}$ | $0.26_{\pm 0.02}$ | $0.01_{\pm 0.01}$ | $0.16_{\pm 0.01}$ | $0.06_{\pm 0.02}$ | $0.55_{\pm 0.04}$ |
| Fisher | $99.51_{\pm 0.17}$ | $3.83_{\pm 0.44}$ | $0.07_{\pm 0.01}$ | $0.02_{\pm 0.00}$ | $1.83_{\pm 0.06}$ | $0.07_{\pm 0.02}$ | $0.56_{\pm 0.04}$ |
| RELOAD | $\mathbf{99.49_{\pm 0.10}}$ | $1.83_{\pm 0.83}$ | $\mathbf{0.05_{\pm 0.04}}$ | $\mathbf{0.00_{\pm 0.00}}$ | $0.26_{\pm 0.09}$ | $0.12_{\pm 0.01}$ | $0.53_{\pm 0.07}$ |

Table 12: **10% Random Forgetting on CIFAR-10 (ResNet-18)**
$\uparrow$: the goal is to have as high of a value as possible, $\Delta^{\downarrow}$: the value in the table is the difference between the result of the unlearning method and retraining (top row) on the metric and the goal is to have a low difference, $\downarrow$: the goal is to have as low of a value as possible. The top row presents the value of $\mathcal{M}^{(\theta^{\sim})}$ on each metric. Subsequent rows for $\Delta$FA ($\downarrow$), $\Delta$FE ($\downarrow$), and $\Delta$FMIA ($\downarrow$) present the absolute difference in the value of the corresponding method on this metric to the value of $\mathcal{M}^{(\theta^{\sim})}$ on the metric. These results show that RELOAD outperforms all the baselines on NA, $\Delta$FE, $\Delta$FMIA by large margins. RELOAD performs competitively on the $\Delta$FA, FSKL, and NSKL metrics, but is outperformed by FT. RELOAD incurs a higher computational cost than other baselines other than FT.

| Method | NA ($\uparrow$) | $\Delta$FA ($\downarrow$) | $\Delta$FE ($\downarrow$) | $\Delta$FMIA ($\downarrow$) | Cost ($\downarrow$) | NSKL ($\downarrow$) | FSKL ($\downarrow$) |
|---|---|---|---|---|---|---|---|
| Retrain | $99.99_{\pm0.00}$ | $95.16_{\pm0.30}$ | $0.20_{\pm0.02}$ | $0.50_{\pm0.00}$ | $1.00_{\pm0.00}$ | $0.00_{\pm0.00}$ | $0.00_{\pm0.00}$ |
| GA | $98.38_{\pm0.21}$ | $4.40_{\pm0.41}$ | $0.18_{\pm0.02}$ | $0.05_{\pm0.01}$ | $\mathbf{0.00}_{\pm\mathbf{0.00}}$ | $0.06_{\pm0.02}$ | $0.66_{\pm0.06}$ |
| FT | $98.24_{\pm0.21}$ | $4.33_{\pm0.37}$ | $0.18_{\pm0.02}$ | $0.04_{\pm0.01}$ | $0.26_{\pm0.02}$ | $\mathbf{0.05}_{\pm\mathbf{0.01}}$ | $0.48_{\pm0.04}$ |
| SSD | $20.02_{\pm29.99}$ | $75.41_{\pm26.74}$ | $1.89_{\pm0.62}$ | $0.02_{\pm0.03}$ | $0.01_{\pm0.00}$ | $8.30_{\pm3.11}$ | $7.83_{\pm2.70}$ |
| SCRUB | $98.41_{\pm0.20}$ | $4.47_{\pm0.40}$ | $0.19_{\pm0.02}$ | $0.05_{\pm0.01}$ | $0.02_{\pm0.00}$ | $0.06_{\pm0.02}$ | $0.65_{\pm0.04}$ |
| CF-$k$ | $98.28_{\pm0.23}$ | $4.47_{\pm0.39}$ | $0.19_{\pm0.02}$ | $0.05_{\pm0.01}$ | $0.17_{\pm0.01}$ | $0.06_{\pm0.02}$ | $0.55_{\pm0.04}$ |
| EU-$k$ | $98.31_{\pm0.21}$ | $4.48_{\pm0.40}$ | $0.19_{\pm0.02}$ | $0.06_{\pm0.01}$ | $0.17_{\pm0.01}$ | $0.07_{\pm0.02}$ | $0.56_{\pm0.04}$ |
| SalUn | $99.86_{\pm0.04}$ | $1.98_{\pm0.48}$ | $0.09_{\pm0.02}$ | $0.04_{\pm0.01}$ | $0.17_{\pm0.01}$ | $0.06_{\pm0.02}$ | $0.55_{\pm0.04}$ |
| Fisher | $99.61_{\pm0.14}$ | $0.15_{\pm0.06}$ | $0.00_{\pm0.00}$ | $0.01_{\pm0.01}$ | $2.17_{\pm0.04}$ | $0.07_{\pm0.02}$ | $0.56_{\pm0.04}$ |
| RELOAD | $\mathbf{99.76}_{\pm\mathbf{0.16}}$ | $\mathbf{0.08}_{\pm\mathbf{0.08}}$ | $\mathbf{0.01}_{\pm\mathbf{0.00}}$ | $\mathbf{0.00}_{\pm\mathbf{0.00}}$ | $0.12_{\pm0.01}$ | $\mathbf{0.05}_{\pm\mathbf{0.03}}$ | $\mathbf{0.19}_{\pm\mathbf{0.02}}$ |

Table 13: **10% Random Forgetting on SVHN (ResNet-18)**
$\uparrow$: the goal is to have as high of a value as possible, $\Delta^{\downarrow}$: the value in the table is the difference between the result of the unlearning method and retraining (top row) on the metric and the goal is to have a low difference, $\downarrow$: the goal is to have as low of a value as possible. The top row presents the value of $\mathcal{M}^{(\theta^{\sim})}$ on each metric. Subsequent rows for $\Delta$FA ($\downarrow$), $\Delta$FE ($\downarrow$), and $\Delta$FMIA ($\downarrow$) present the absolute difference in the value of the corresponding method on this metric to the value of $\mathcal{M}^{(\theta^{\sim})}$ on the metric. These results show that RELOAD outperforms all the baselines on NA, $\Delta$FA, $\Delta$FE, $\Delta$FMIA, FSKL, and NSKL by large margins. RELOAD performs competitively on the Cost, but incurs a higher computational cost than other baselines other than FT, CF-$k$, EU-$k$.

| Method | NA ($\uparrow$) | $\Delta$FA ($\downarrow$) | $\Delta$FE ($\downarrow$) | $\Delta$FMIA ($\downarrow$) | Cost ($\downarrow$) | NSKL ($\downarrow$) | FSKL ($\downarrow$) |
|---|---|---|---|---|---|---|---|
| Retrain | $99.99_{\pm0.00}$ | $95.08_{\pm0.31}$ | $0.24_{\pm0.02}$ | $0.50_{\pm0.00}$ | $1.00_{\pm0.00}$ | $0.00_{\pm0.00}$ | $0.00_{\pm0.00}$ |
| GA | $98.40_{\pm0.23}$ | $4.43_{\pm0.44}$ | $0.21_{\pm0.02}$ | $0.03_{\pm0.01}$ | $\mathbf{0.00}_{\pm\mathbf{0.00}}$ | $0.06_{\pm0.02}$ | $0.65_{\pm0.06}$ |
| FT | $98.30_{\pm0.18}$ | $4.49_{\pm0.43}$ | $0.22_{\pm0.02}$ | $0.03_{\pm0.01}$ | $0.24_{\pm0.03}$ | $\mathbf{0.05}_{\pm\mathbf{0.01}}$ | $\mathbf{0.49}_{\pm\mathbf{0.04}}$ |
| SSD | $22.88_{\pm34.01}$ | $70.45_{\pm29.04}$ | $1.80_{\pm0.69}$ | $0.01_{\pm0.01}$ | $\mathbf{0.00}_{\pm\mathbf{0.00}}$ | $7.99_{\pm3.52}$ | $7.56_{\pm3.06}$ |
| SCRUB | $98.43_{\pm0.22}$ | $4.50_{\pm0.41}$ | $0.22_{\pm0.02}$ | $0.03_{\pm0.01}$ | $0.02_{\pm0.00}$ | $0.06_{\pm0.02}$ | $0.66_{\pm0.04}$ |
| CF-$k$ | $98.34_{\pm0.24}$ | $4.51_{\pm0.42}$ | $0.22_{\pm0.02}$ | $0.04_{\pm0.01}$ | $0.21_{\pm0.03}$ | $0.06_{\pm0.02}$ | $0.55_{\pm0.05}$ |
| EU-$k$ | $98.34_{\pm0.23}$ | $4.51_{\pm0.42}$ | $0.22_{\pm0.02}$ | $0.04_{\pm0.01}$ | $0.21_{\pm0.03}$ | $0.06_{\pm0.02}$ | $0.56_{\pm0.05}$ |
| SalUn | $99.94_{\pm0.02}$ | $3.88_{\pm0.62}$ | $0.13_{\pm0.01}$ | $0.04_{\pm0.01}$ | $0.15_{\pm0.00}$ | $0.06_{\pm0.02}$ | $0.55_{\pm0.05}$ |
| Fisher | $99.55_{\pm0.18}$ | $0.04_{\pm0.04}$ | $0.00_{\pm0.00}$ | $0.00_{\pm0.00}$ | $1.46_{\pm0.03}$ | $0.06_{\pm0.02}$ | $0.56_{\pm0.05}$ |
| RELOAD | $\mathbf{99.50}_{\pm\mathbf{0.11}}$ | $\mathbf{0.65}_{\pm\mathbf{0.72}}$ | $\mathbf{0.04}_{\pm\mathbf{0.04}}$ | $\mathbf{0.00}_{\pm\mathbf{0.00}}$ | $0.26_{\pm0.10}$ | $0.12_{\pm0.01}$ | $0.53_{\pm0.08}$ |

Table 14: **10% Random Forgetting on SVHN (VGG16-BN)**
$\uparrow$: the goal is to have as high of a value as possible, $\Delta^{\downarrow}$: the value in the table is the difference between the result of the unlearning method and retraining (top row) on the metric and the goal is to have a low difference, $\downarrow$: the goal is to have as low of a value as possible. The top row presents the value of $\mathcal{M}^{(\theta^{\sim})}$ on each metric. Subsequent rows for $\Delta$FA ($\downarrow$), $\Delta$FE ($\downarrow$), and $\Delta$FMIA ($\downarrow$) present the absolute difference in the value of the corresponding method on this metric to the value of $\mathcal{M}^{(\theta^{\sim})}$ on the metric. These results show that RELOAD outperforms all the baselines on NA, $\Delta$FA, $\Delta$FE, and $\Delta$FMIA, by large margins. RELOAD performs competitively on NSKL and FSKL but is outperformed by FT. RELOAD also incurs a higher computational cost than the other baselines.

| Method | NA (↑) | ΔFA (↓) | ΔFE (↓) | ΔFMIA (↓) | Cost (↓) | NSKL (↓) | FSKL (↓) |
|---|---|---|---|---|---|---|---|
| Retrain | $97.80_{\pm 0.33}$ | $68.25_{\pm 0.49}$ | $1.82_{\pm 0.06}$ | $0.50_{\pm 0.01}$ | $1.00_{\pm 0.00}$ | $0.00_{\pm 0.00}$ | $0.00_{\pm 0.00}$ |
| GA | $98.41_{\pm 0.25}$ | $26.40_{\pm 1.18}$ | $1.64_{\pm 0.07}$ | $0.14_{\pm 0.03}$ | $\mathbf{0.00}_{\pm 0.00}$ | $\mathbf{0.06}_{\pm 0.03}$ | $0.66_{\pm 0.06}$ |
| FT | $98.27_{\pm 0.20}$ | $12.65_{\pm 1.81}$ | $1.16_{\pm 0.07}$ | $0.08_{\pm 0.02}$ | $0.25_{\pm 0.03}$ | $\mathbf{0.06}_{\pm 0.01}$ | $\mathbf{0.50}_{\pm 0.03}$ |
| SSD | $22.86_{\pm 34.01}$ | $61.38_{\pm 15.72}$ | $2.57_{\pm 0.55}$ | $0.02_{\pm 0.05}$ | $\mathbf{0.00}_{\pm 0.00}$ | $8.01_{\pm 3.53}$ | $7.57_{\pm 3.07}$ |
| SCRUB | $98.43_{\pm 0.23}$ | $26.62_{\pm 1.10}$ | $1.66_{\pm 0.06}$ | $0.14_{\pm 0.03}$ | $0.02_{\pm 0.00}$ | $\mathbf{0.06}_{\pm 0.02}$ | $0.66_{\pm 0.04}$ |
| CF-$k$ | $98.30_{\pm 0.27}$ | $26.26_{\pm 1.25}$ | $1.68_{\pm 0.06}$ | $0.15_{\pm 0.02}$ | $0.27_{\pm 0.04}$ | $\mathbf{0.06}_{\pm 0.02}$ | $0.56_{\pm 0.04}$ |
| EU-$k$ | $98.35_{\pm 0.25}$ | $26.16_{\pm 1.26}$ | $1.67_{\pm 0.06}$ | $0.15_{\pm 0.02}$ | $0.27_{\pm 0.04}$ | $\mathbf{0.06}_{\pm 0.02}$ | $0.57_{\pm 0.04}$ |
| RELOAD | $\mathbf{99.51}_{\pm 0.09}$ | $\mathbf{3.37}_{\pm 1.55}$ | $\mathbf{0.40}_{\pm 0.07}$ | $\mathbf{0.02}_{\pm 0.01}$ | $0.24_{\pm 0.11}$ | $0.11_{\pm 0.01}$ | $0.51_{\pm 0.03}$ |

Table 15: **10% Random Forgetting on CIFAR-100(VGG16-BN)**
↑: the goal is to have as high of a value as possible, $\Delta^{\downarrow}$: the value in the table is the difference between the result of the unlearning method and retraining (top row) on the metric and the goal is to have a low difference, ↓: the goal is to have as low of a value as possible. The top row presents the value of $\mathcal{M}^{(\theta^{\sim})}$ on each metric. Subsequent rows for ΔFA (↓), ΔFE (↓), and ΔFMIA (↓) present the absolute difference in the value of the corresponding method on this metric to the value of $\mathcal{M}^{(\theta^{\sim})}$ on the metric. These results show that RELOAD outperforms all the baselines on NA, ΔFA, ΔFE, and ΔFMIA, by large margins. RELOAD performs competitively on NSKL and FSKL but is outperformed by FT. RELOAD also incurs a higher computational cost than other baselines other than FT, CF-$k$, and EU-$k$.

## A.7 RANDOM 30% FORGETTING

| Method | NA (↑) | ΔFA (↓) | ΔFE (↓) | ΔFMIA (↓) | ΔAUC (↓) | Cost (↓) | NSKL (↓) | FSKL (↓) |
|---|---|---|---|---|---|---|---|---|

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

Table 23: **100 In Class Random Forgetting on CIFAR-10 (VGG16-BN).** The top row presents the value of $\mathcal{M}^{(\theta^\sim)}$ on each metric. Subsequent rows for $\Delta$FA ($\downarrow$), $\Delta$FE ($\downarrow$), and $\Delta$FMIA ($\downarrow$) present the absolute difference in the value of the corresponding method on this metric to the value of $\mathcal{M}^{(\theta^\sim)}$ on the metric. These results show that RELOAD outperforms all baselines on $\Delta$FA, $\Delta$FE, $\Delta$FMIA, FSKL indicating it behaves the closes to $\mathcal{M}^{(\theta^\sim)}$ on $\mathcal{D}_{forget}$. RELOAD performs competitively on NA and NSKL, falling behind of the leading method by 0.46 for NA and 0.02 for NSKL.

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

## A.9 CROSS PATTERN BACKDOOR ATTACK REMEDIATION - ADDITIONAL EXPERIMENTS

**Metrics.** In Table 7.

| Method | NA ($\uparrow$) | TNA ($\uparrow$) | OA ($\uparrow$) | TA ($\uparrow$) | TTA ($\uparrow$) | Cost ($\downarrow$) |
|---|---|---|---|---|---|---|
| Original | $87.46_{\pm 0.56}$ | $20.00_{\pm 0.00}$ | $98.87_{\pm 0.14}$ | $82.68_{\pm 0.45}$ | $19.81_{\pm 0.03}$ | N/A |
| Retrain | $98.98_{\pm 0.04}$ | $98.53_{\pm 0.02}$ | $98.88_{\pm 0.02}$ | $92.48_{\pm 0.00}$ | $91.90_{\pm 0.00}$ | $1.00_{\pm 0.00}$ |
| GAR | $60.18_{\pm 37.89}$ | $59.34_{\pm 37.09}$ | $61.52_{\pm 39.32}$ | $57.29_{\pm 34.88}$ | $56.54_{\pm 34.12}$ | $0.08_{\pm 0.01}$ |
| GRDA | $65.82_{\pm 31.39}$ | $65.16_{\pm 30.64}$ | $66.90_{\pm 32.50}$ | $62.87_{\pm 28.47}$ | $62.34_{\pm 27.80}$ | $0.05_{\pm 0.00}$ |
| FT | $94.36_{\pm 4.34}$ | $93.90_{\pm 4.07}$ | $94.42_{\pm 4.34}$ | $86.87_{\pm 4.39}$ | $86.50_{\pm 4.14}$ | $0.37_{\pm 0.02}$ |
| SSD | $30.71_{\pm 24.06}$ | $23.93_{\pm 13.18}$ | $31.78_{\pm 26.91}$ | $30.25_{\pm 22.90}$ | $23.94_{\pm 13.43}$ | $\mathbf{0.01}_{\pm 0.00}$ |
| SCRUB | $12.44_{\pm 3.50}$ | $12.44_{\pm 3.51}$ | $12.44_{\pm 3.50}$ | $12.43_{\pm 3.45}$ | $12.42_{\pm 3.44}$ | $0.04_{\pm 0.01}$ |
| CF-$k$ | $70.01_{\pm 27.07}$ | $69.62_{\pm 26.66}$ | $70.28_{\pm 27.40}$ | $66.56_{\pm 24.27}$ | $66.29_{\pm 23.80}$ | $0.29_{\pm 0.03}$ |
| EU-$k$ | $70.10_{\pm 27.00}$ | $69.71_{\pm 26.60}$ | $70.34_{\pm 27.34}$ | $66.75_{\pm 24.08}$ | $66.41_{\pm 23.63}$ | $0.29_{\pm 0.03}$ |
| RELOAD | $\mathbf{99.89}_{\pm 0.10}$ | $\mathbf{99.62}_{\pm 0.45}$ | $\mathbf{99.65}_{\pm 0.43}$ | $\mathbf{90.81}_{\pm 0.99}$ | $\mathbf{90.51}_{\pm 0.82}$ | $0.08_{\pm 0.06}$ |

Table 27: **Cross Pattern Backdoor Attack on CIFAR-10 (ResNet-18).** $\uparrow$: the goal is to have as high of a value as possible, $\downarrow$: the goal is to have as low of a value as possible. These results show that RELOAD outperforms all baselines on NA, TNA, OA, TA, and TTA. The small differences between the values of NA and TNA, and TA and TTA for RELOAD indicate that it successfully removed the influence of an injected backdoor. RELOAD incurs a higher computational cost than most baselines, but is cheaper than FT, CF-$k$, and EU-$k$.

| Method | NA ($\uparrow$) | TNA ($\uparrow$) | OA ($\uparrow$) | TA ($\uparrow$) | TTA ($\uparrow$) | Cost ($\downarrow$) |
|---|---|---|---|---|---|---|
| Original | $87.46_{\pm 0.56}$ | $20.00_{\pm 0.00}$ | $98.87_{\pm 0.14}$ | $82.68_{\pm 0.45}$ | $19.81_{\pm 0.03}$ | N/A |
| Retrain | $99.69_{\pm 0.02}$ | $99.29_{\pm 0.02}$ | $99.59_{\pm 0.02}$ | $92.39_{\pm 0.00}$ | $91.50_{\pm 0.00}$ | $1.00_{\pm 0.00}$ |
| GAR | $96.32_{\pm 0.41}$ | $94.78_{\pm 0.52}$ | $99.11_{\pm 0.15}$ | $89.15_{\pm 0.20}$ | $87.31_{\pm 0.22}$ | $0.08_{\pm 0.01}$ |
| GRDA | $96.06_{\pm 0.25}$ | $94.16_{\pm 0.27}$ | $97.83_{\pm 0.43}$ | $88.94_{\pm 0.14}$ | $86.73_{\pm 0.16}$ | $0.05_{\pm 0.00}$ |
| FT | $98.38_{\pm 0.28}$ | $97.54_{\pm 0.37}$ | $98.47_{\pm 0.26}$ | $90.17_{\pm 0.42}$ | $89.28_{\pm 0.47}$ | $0.41_{\pm 0.01}$ |
| SSD | $30.71_{\pm 24.06}$ | $23.93_{\pm 13.18}$ | $31.78_{\pm 26.91}$ | $30.25_{\pm 22.90}$ | $23.94_{\pm 13.43}$ | $\mathbf{0.01}_{\pm 0.00}$ |
| SCRUB | $12.44_{\pm 3.50}$ | $12.44_{\pm 3.51}$ | $12.44_{\pm 3.50}$ | $12.43_{\pm 3.45}$ | $12.42_{\pm 3.44}$ | $0.06_{\pm 0.01}$ |
| CF-$k$ | $70.01_{\pm 27.07}$ | $69.62_{\pm 26.66}$ | $70.28_{\pm 27.40}$ | $66.56_{\pm 24.27}$ | $66.29_{\pm 23.80}$ | $0.53_{\pm 0.09}$ |
| EU-$k$ | $70.10_{\pm 27.00}$ | $69.71_{\pm 26.60}$ | $70.34_{\pm 27.34}$ | $66.75_{\pm 24.08}$ | $66.41_{\pm 23.63}$ | $0.53_{\pm 0.09}$ |
| RELOAD | $\mathbf{99.89}_{\pm 0.10}$ | $\mathbf{99.62}_{\pm 0.45}$ | $\mathbf{99.65}_{\pm 0.43}$ | $\mathbf{90.81}_{\pm 0.99}$ | $\mathbf{90.51}_{\pm 0.82}$ | $0.12_{\pm 0.07}$ |

Table 28: **Cross Pattern Backdoor Attack on CIFAR-10(VGG16-BN)**
$\uparrow$: the goal is to have as high of a value as possible, $\downarrow$: the goal is to have as low of a value as possible. The top row presents the values of $\mathcal{M}^{(\theta^*)}$ on these metrics. These results show that RELOAD outperforms all the baselines on NA, TNA, OA, TA, and TTA. RELOAD incurs a higher computational cost than the other baselines.

| Method | NA ($\uparrow$) | TNA ($\uparrow$) | OA ($\uparrow$) | TA ($\uparrow$) | TTA ($\uparrow$) | Cost ($\downarrow$) |
|---|---|---|---|---|---|---|
| Original | $81.42_{\pm 0.48}$ | $19.88_{\pm 0.09}$ | $98.77_{\pm 0.79}$ | $60.48_{\pm 0.32}$ | $16.91_{\pm 0.17}$ | N/A |
| Retrain | $97.21_{\pm 0.03}$ | $95.11_{\pm 0.05}$ | $96.61_{\pm 0.03}$ | $68.41_{\pm 0.00}$ | $66.25_{\pm 0.00}$ | $1.00_{\pm 0.00}$ |
| GAR | $90.31_{\pm 1.72}$ | $85.45_{\pm 2.94}$ | $95.50_{\pm 1.09}$ | $\mathbf{65.48}_{\pm 0.52}$ | $62.06_{\pm 1.42}$ | $0.09_{\pm 0.01}$ |
| GRDA | $86.21_{\pm 1.73}$ | $81.59_{\pm 2.43}$ | $88.68_{\pm 2.06}$ | $62.92_{\pm 1.00}$ | $59.96_{\pm 1.45}$ | $0.05_{\pm 0.01}$ |
| FT | $93.60_{\pm 0.88}$ | $91.29_{\pm 1.15}$ | $93.86_{\pm 1.07}$ | $65.07_{\pm 0.59}$ | $\mathbf{63.27}_{\pm 0.68}$ | $0.44_{\pm 0.06}$ |
| SSD | $9.12_{\pm 25.69}$ | $2.79_{\pm 5.65}$ | $10.73_{\pm 30.76}$ | $7.01_{\pm 19.02}$ | $2.48_{\pm 4.71}$ | $\mathbf{0.01}_{\pm 0.00}$ |
| SCRUB | $78.63_{\pm 0.94}$ | $81.66_{\pm 2.27}$ | $97.41_{\pm 0.98}$ | $57.37_{\pm 0.55}$ | $59.50_{\pm 1.06}$ | $0.06_{\pm 0.01}$ |
| CF-$k$ | $90.08_{\pm 1.17}$ | $86.67_{\pm 1.41}$ | $93.46_{\pm 1.13}$ | $64.66_{\pm 0.56}$ | $62.16_{\pm 0.88}$ | $0.49_{\pm 0.07}$ |
| EU-$k$ | $90.07_{\pm 1.15}$ | $86.44_{\pm 1.41}$ | $93.37_{\pm 1.09}$ | $64.61_{\pm 0.56}$ | $61.96_{\pm 0.92}$ | $0.48_{\pm 0.08}$ |
| RELOAD | $\mathbf{99.84}_{\pm 0.14}$ | $\mathbf{99.35}_{\pm 0.59}$ | $\mathbf{99.77}_{\pm 0.19}$ | $59.37_{\pm 6.07}$ | $59.12_{\pm 6.04}$ | $0.21_{\pm 0.09}$ |

Table 29: **Cross Pattern Backdoor Attack on CIFAR-100(VGG16-BN)**
$\uparrow$: the goal is to have as high of a value as possible, $\downarrow$: the goal is to have as low of a value as possible. The top row presents the values of $\mathcal{M}^{(\theta^*)}$ on these metrics. These results show that RELOAD outperforms all the baselines on NA, TNA, and OA. RELOAD performs competitively on TA and TTA but is outperformed. RELOAD also incurs a higher computational cost than the other baselines.

| Method | NA ($\uparrow$) | TNA ($\uparrow$) | OA ($\uparrow$) | TA ($\uparrow$) | TTA ($\uparrow$) | Cost ($\downarrow$) |
|---|---|---|---|---|---|---|
| Original | $85.50_{\pm 0.37}$ | $25.41_{\pm 9.03}$ | $98.46_{\pm 0.20}$ | $64.96_{\pm 0.37}$ | $21.17_{\pm 5.93}$ | N/A |
| Retrain | $99.69_{\pm 0.02}$ | $99.29_{\pm 0.02}$ | $99.59_{\pm 0.02}$ | $92.39_{\pm 0.00}$ | $91.50_{\pm 0.00}$ | $1.00_{\pm 0.00}$ |
| GAR | $96.32_{\pm 0.41}$ | $94.78_{\pm 0.52}$ | $99.11_{\pm 0.15}$ | $89.15_{\pm 0.20}$ | $87.31_{\pm 0.22}$ | $0.05_{\pm 0.00}$ |
| GRDA | $96.06_{\pm 0.25}$ | $94.16_{\pm 0.27}$ | $97.83_{\pm 0.43}$ | $88.94_{\pm 0.14}$ | $86.73_{\pm 0.16}$ | $0.03_{\pm 0.00}$ |
| FT | $98.38_{\pm 0.28}$ | $97.54_{\pm 0.37}$ | $98.47_{\pm 0.26}$ | $\mathbf{90.17}_{\pm 0.42}$ | $\mathbf{89.28}_{\pm 0.47}$ | $0.22_{\pm 0.00}$ |
| SSD | $17.46_{\pm 23.58}$ | $11.00_{\pm 3.17}$ | $18.90_{\pm 28.14}$ | $16.92_{\pm 21.76}$ | $11.02_{\pm 3.11}$ | $\mathbf{0.00}_{\pm 0.00}$ |
| SCRUB | $81.58_{\pm 0.30}$ | $92.06_{\pm 0.15}$ | $99.53_{\pm 0.24}$ | $75.25_{\pm 0.46}$ | $84.89_{\pm 0.13}$ | $0.03_{\pm 0.00}$ |
| CF-$k$ | $92.99_{\pm 0.34}$ | $90.96_{\pm 0.77}$ | $97.34_{\pm 0.29}$ | $85.21_{\pm 0.24}$ | $83.29_{\pm 0.77}$ | $0.24_{\pm 0.00}$ |
| EU-$k$ | $93.03_{\pm 0.35}$ | $91.06_{\pm 0.87}$ | $97.43_{\pm 0.26}$ | $85.31_{\pm 0.29}$ | $83.31_{\pm 0.79}$ | $0.24_{\pm 0.00}$ |
| RELOAD | $\mathbf{99.87}_{\pm 0.06}$ | $\mathbf{99.36}_{\pm 0.83}$ | $\mathbf{99.72}_{\pm 0.28}$ | $72.45_{\pm 0.91}$ | $71.80_{\pm 0.96}$ | $0.03_{\pm 0.01}$ |

Table 30: **Cross Pattern Backdoor Attack on CIFAR-100(ResNet-18)**
$\uparrow$: the goal is to have as high of a value as possible, $\downarrow$: the goal is to have as low of a value as possible. The top row presents the values of $\mathcal{M}^{(\theta^*)}$ on these metrics. These results show that RELOAD outperforms all the baselines on NA, TNA, and OA. RELOAD performs competitively on TA and TTA but is outperformed by FT. RELOAD also incurs a competitive computational cost and is only more expensive than SSD.

| Method | NA ($\uparrow$) | TNA ($\uparrow$) | OA ($\uparrow$) | TA ($\uparrow$) | TTA ($\uparrow$) | Cost ($\downarrow$) |
|---|---|---|---|---|---|---|
| Original | $76.01_{\pm 4.06}$ | $24.29_{\pm 1.93}$ | $93.59_{\pm 5.42}$ | $72.66_{\pm 4.40}$ | $24.98_{\pm 1.90}$ | N/A |
| Retrain | $99.99_{\pm 0.00}$ | $99.99_{\pm 0.00}$ | $99.99_{\pm 0.00}$ | $95.45_{\pm 0.00}$ | $95.41_{\pm 0.00}$ | $1.00_{\pm 0.00}$ |
| GAR | $81.74_{\pm 36.14}$ | $76.10_{\pm 33.27}$ | $82.44_{\pm 36.52}$ | $77.83_{\pm 33.89}$ | $72.77_{\pm 31.29}$ | $0.08_{\pm 0.00}$ |
| GRDA | $34.08_{\pm 34.81}$ | $31.40_{\pm 34.41}$ | $31.87_{\pm 35.31}$ | $33.38_{\pm 33.19}$ | $30.92_{\pm 32.79}$ | $0.05_{\pm 0.00}$ |
| FT | $\mathbf{99.70}_{\pm 0.82}$ | $\mathbf{99.50}_{\pm 0.94}$ | $\mathbf{99.69}_{\pm 0.82}$ | $\mathbf{94.60}_{\pm 0.71}$ | $\mathbf{94.31}_{\pm 0.89}$ | $0.40_{\pm 0.01}$ |
| SSD | $18.21_{\pm 19.48}$ | $12.76_{\pm 2.80}$ | $19.90_{\pm 24.81}$ | $17.96_{\pm 18.32}$ | $12.97_{\pm 3.55}$ | $\mathbf{0.01}_{\pm 0.00}$ |
| SCRUB | $6.75_{\pm 0.00}$ | $6.76_{\pm 0.00}$ | $6.76_{\pm 0.00}$ | $6.71_{\pm 0.00}$ | $6.71_{\pm 0.00}$ | $0.05_{\pm 0.00}$ |
| CF-$k$ | $93.19_{\pm 2.46}$ | $90.95_{\pm 7.44}$ | $97.24_{\pm 4.70}$ | $87.31_{\pm 1.90}$ | $85.08_{\pm 5.93}$ | $0.37_{\pm 0.01}$ |
| EU-$k$ | $93.26_{\pm 2.49}$ | $91.05_{\pm 7.58}$ | $97.09_{\pm 4.94}$ | $87.39_{\pm 1.86}$ | $85.16_{\pm 5.97}$ | $0.37_{\pm 0.01}$ |
| RELOAD | $99.21_{\pm 0.94}$ | $98.62_{\pm 2.66}$ | $99.10_{\pm 1.26}$ | $94.32_{\pm 3.05}$ | $94.14_{\pm 3.61}$ | $0.28_{\pm 0.18}$ |

Table 31: **Cross Pattern Backdoor Attack on SVHN (VGG16-BN)**
$\uparrow$: the goal is to have as high of a value as possible, $\downarrow$: the goal is to have as low of a value as possible. The top row presents the values of $\mathcal{M}^{(\theta^*)}$ on these metrics. These results show that FT outperforms RELOAD and other baselines on all metrics but is very computationally expensive. RELOAD performs competitively on all metrics, and is narrowly outperformed by FT while providing a significantly lower computational cost.

| Method | NA ($\uparrow$) | TNA ($\uparrow$) | OA ($\uparrow$) | TA ($\uparrow$) | TTA ($\uparrow$) | Cost ($\downarrow$) |
|---|---|---|---|---|---|---|
| Original | $75.29_{\pm 0.68}$ | $23.92_{\pm 2.31}$ | $96.12_{\pm 2.45}$ | $71.35_{\pm 0.43}$ | $24.38_{\pm 2.06}$ | N/A |
| Retrain | $100.00_{\pm 0.00}$ | $100.00_{\pm 0.00}$ | $99.99_{\pm 0.00}$ | $95.26_{\pm 0.00}$ | $95.22_{\pm 0.00}$ | $1.00_{\pm 0.00}$ |
| GAR | $99.36_{\pm 0.04}$ | $95.62_{\pm 1.02}$ | $100.00_{\pm 0.00}$ | $94.35_{\pm 0.15}$ | $90.40_{\pm 1.06}$ | $0.07_{\pm 0.00}$ |
| GRDA | $96.60_{\pm 2.49}$ | $47.19_{\pm 37.23}$ | $49.87_{\pm 44.15}$ | $91.15_{\pm 3.03}$ | $44.57_{\pm 35.54}$ | $0.05_{\pm 0.00}$ |
| FT | $\mathbf{100.00}_{\pm 0.00}$ | $\mathbf{100.00}_{\pm 0.00}$ | $\mathbf{100.00}_{\pm 0.00}$ | $\mathbf{95.30}_{\pm 0.13}$ | $\mathbf{95.24}_{\pm 0.15}$ | $0.36_{\pm 0.01}$ |
| SSD | $19.65_{\pm 20.04}$ | $14.57_{\pm 4.22}$ | $21.80_{\pm 26.80}$ | $19.96_{\pm 18.52}$ | $15.33_{\pm 4.49}$ | $\mathbf{0.01}_{\pm 0.00}$ |
| SCRUB | $23.42_{\pm 7.68}$ | $23.40_{\pm 7.69}$ | $23.35_{\pm 7.70}$ | $24.42_{\pm 8.17}$ | $24.43_{\pm 8.17}$ | $0.05_{\pm 0.00}$ |
| CF-$k$ | $99.19_{\pm 0.12}$ | $97.21_{\pm 0.42}$ | $99.59_{\pm 0.22}$ | $93.14_{\pm 0.16}$ | $90.59_{\pm 0.40}$ | $0.25_{\pm 0.00}$ |
| EU-$k$ | $99.19_{\pm 0.11}$ | $97.02_{\pm 0.42}$ | $99.57_{\pm 0.22}$ | $93.12_{\pm 0.14}$ | $90.32_{\pm 0.37}$ | $0.24_{\pm 0.00}$ |
| RELOAD | $99.57_{\pm 0.16}$ | $99.54_{\pm 0.18}$ | $99.57_{\pm 0.17}$ | $94.69_{\pm 0.78}$ | $94.68_{\pm 0.77}$ | $0.13_{\pm 0.05}$ |

Table 32: **Cross Pattern Backdoor Attack on SVHN (ResNet-18)**
$\uparrow$: the goal is to have as high of a value as possible, $\downarrow$: the goal is to have as low of a value as possible. The top row presents the values of $\mathcal{M}^{(\theta^*)}$ on these metrics. These results show that FT outperforms RELOAD and other baselines on all metrics but is very computationally expensive. RELOAD performs competitively on all metrics, and is narrowly outperformed by FT while providing a significantly lower computational cost.

## A.10 TARGETED MISLABELLING CORRECTION - ADDITIONAL EXPERIMENTS

**Label Flip Attack.** In this setting, we select 1 class from $\mathcal{D}$ and selectively change the labels of its training samples to construct $\mathcal{D}$. The specific class pairings are detailed in Appendix A.3. Using the original dataset with the correct labels as $\mathcal{D}_{new}$, we evaluate RELOAD and the other baselines on correcting the mislabelling. The results of this experiment are shown in Appendix

A.10. Observe that the effect of this label flip attack produces a trained model (Original) which has degraded performance on NA and TA. Notice that RELOAD successfully remedies the effects of this attack, achieving the 2nd highest evaluations across the board on NA, OA, and TA suggesting that RELOAD. Naive fine-tuning however, outperforms on this task, achieving the best results by a small margin on NA, OA and TA. It is likely that due to the uniformity of the label attack (all mislabelling being from a single class to another single semantically similar class), FT was able to be sufficient in correcting it. Potential future work may explore more complex patterns of mislabelling. Additionally, RELOAD is more than twice as computationally efficient as fine-tuning. Both Fine-tuning, and RELOAD produce models which bear significantly similar results to that of the retrained model, on all metrics, implying the correction was sufficiently accomplished.

| Method | NA ($\uparrow$) | OA ($\uparrow$) | TA ($\uparrow$) | Cost ($\downarrow$) |
|---|---|---|---|---|
| Original | $91.06_{\pm 1.92}$ | $99.99_{\pm 0.00}$ | $87.27_{\pm 1.78}$ | N/A |
| Retrain | $99.99_{\pm 0.00}$ | $91.06_{\pm 1.92}$ | $95.26_{\pm 0.00}$ | $1.00_{\pm 0.00}$ |
| GAR | $91.06_{\pm 1.92}$ | $98.17_{\pm 1.88}$ | $87.29_{\pm 1.75}$ | $0.07_{\pm 0.00}$ |
| GRDA | $17.70_{\pm 2.92}$ | $10.47_{\pm 5.23}$ | $16.61_{\pm 3.67}$ | $0.05_{\pm 0.00}$ |
| FT | $\mathbf{99.99_{\pm 0.00}}$ | $\mathbf{91.06_{\pm 1.92}}$ | $\mathbf{95.48_{\pm 0.15}}$ | $0.35_{\pm 0.02}$ |
| SSD | $74.47_{\pm 12.31}$ | $75.23_{\pm 13.70}$ | $71.48_{\pm 12.48}$ | $\mathbf{0.01_{\pm 0.00}}$ |
| SCRUB | $20.52_{\pm 5.78}$ | $21.40_{\pm 9.67}$ | $20.45_{\pm 6.07}$ | $0.05_{\pm 0.00}$ |
| CF-$k$ | $99.43_{\pm 0.28}$ | $91.01_{\pm 2.05}$ | $94.84_{\pm 0.34}$ | $0.24_{\pm 0.00}$ |
| EU-$k$ | $99.44_{\pm 0.27}$ | $91.02_{\pm 2.03}$ | $94.83_{\pm 0.35}$ | $0.24_{\pm 0.00}$ |
| RELOAD | $99.72_{\pm 0.17}$ | $90.82_{\pm 2.00}$ | $95.20_{\pm 0.18}$ | $0.15_{\pm 0.01}$ |

Table 33: **Label Flip Attack on SVHN (ResNet-18)**
$\uparrow$: the goal is to have as high of a value as possible, $\downarrow$: the goal is to have as low of a value as possible. The top row presents the values of $\mathcal{M}^{(\theta^*)}$ on these metrics. These results show that FT outperforms RELOAD and other baselines on all metrics but is very computationally expensive. RELOAD performs competitively on all metrics, and is narrowly outperformed by FT while providing a significantly lower computational cost.

| Method | NA ($\uparrow$) | OA ($\uparrow$) | TA ($\uparrow$) | Cost ($\downarrow$) |
|---|---|---|---|---|
| Original | $89.03_{\pm 0.13}$ | $98.92_{\pm 0.14}$ | $83.88_{\pm 0.59}$ | N/A |
| Retrain | $99.69_{\pm 0.02}$ | $89.76_{\pm 0.03}$ | $92.39_{\pm 0.00}$ | $1.00_{\pm 0.00}$ |
| GAR | $89.65_{\pm 0.10}$ | $\mathbf{99.42_{\pm 0.14}}$ | $83.38_{\pm 0.75}$ | $0.04_{\pm 0.00}$ |
| GRDA | $79.72_{\pm 2.41}$ | $79.73_{\pm 2.43}$ | $74.05_{\pm 2.52}$ | $0.03_{\pm 0.00}$ |
| FT | $98.37_{\pm 0.28}$ | $89.03_{\pm 0.45}$ | $\mathbf{90.18_{\pm 0.52}}$ | $0.22_{\pm 0.00}$ |
| SSD | $85.27_{\pm 8.86}$ | $92.64_{\pm 12.82}$ | $80.76_{\pm 8.12}$ | $\mathbf{0.01_{\pm 0.00}}$ |
| SCRUB | $13.11_{\pm 2.75}$ | $13.14_{\pm 5.72}$ | $13.02_{\pm 2.82}$ | $0.04_{\pm 0.00}$ |
| CF-$k$ | $95.94_{\pm 1.21}$ | $88.53_{\pm 1.17}$ | $90.43_{\pm 0.78}$ | $0.30_{\pm 0.03}$ |
| EU-$k$ | $95.97_{\pm 1.19}$ | $88.68_{\pm 1.12}$ | $90.45_{\pm 0.76}$ | $0.30_{\pm 0.03}$ |
| RELOAD | $\mathbf{99.89_{\pm 0.05}}$ | $89.92_{\pm 0.04}$ | $86.77_{\pm 0.73}$ | $0.13_{\pm 0.01}$ |

Table 34: **Label Flip Attack on CIFAR-10(ResNet-18)**
$\uparrow$: the goal is to have as high of a value as possible, $\downarrow$: the goal is to have as low of a value as possible. The top row presents the values of $\mathcal{M}^{(\theta^*)}$ on these metrics. These results show that RELOAD outperforms all baselines on NA, and performs competitively on OA and TA on which it is outperformed by GAR and FT respectively. RELOAD incurs a higher computational cost than most baselines, but is faster than FT, CF-$k$, and EU-$k$.

| Method | NA ($\uparrow$) | OA ($\uparrow$) | TA ($\uparrow$) | Cost ($\downarrow$) |
|---|---|---|---|---|
| Original | $89.71_{\pm 0.08}$ | $99.69_{\pm 0.08}$ | $83.35_{\pm 0.68}$ | N/A |
| Retrain | $98.97_{\pm 0.03}$ | $89.15_{\pm 0.09}$ | $92.48_{\pm 0.00}$ | $1.00_{\pm 0.00}$ |
| GAR | $88.80_{\pm 0.19}$ | $\mathbf{98.02_{\pm 0.55}}$ | $83.63_{\pm 0.60}$ | $0.14_{\pm 0.01}$ |
| GRDA | $58.73_{\pm 19.24}$ | $58.97_{\pm 19.15}$ | $53.92_{\pm 17.50}$ | $0.10_{\pm 0.00}$ |
| FT | $97.92_{\pm 0.34}$ | $88.45_{\pm 0.31}$ | $\mathbf{90.91_{\pm 0.50}}$ | $0.69_{\pm 0.05}$ |
| SSD | $79.87_{\pm 16.52}$ | $87.85_{\pm 20.46}$ | $75.08_{\pm 15.00}$ | $\mathbf{0.01_{\pm 0.00}}$ |
| SCRUB | $10.00_{\pm 0.00}$ | $9.00_{\pm 3.16}$ | $10.06_{\pm 0.00}$ | $0.05_{\pm 0.00}$ |
| CF-$k$ | $91.55_{\pm 1.34}$ | $89.71_{\pm 6.09}$ | $85.23_{\pm 1.27}$ | $0.47_{\pm 0.06}$ |
| EU-$k$ | $91.44_{\pm 1.21}$ | $89.39_{\pm 6.15}$ | $85.20_{\pm 1.21}$ | $0.47_{\pm 0.06}$ |
| RELOAD | $\mathbf{98.84_{\pm 1.30}}$ | $89.56_{\pm 0.40}$ | $75.80_{\pm 20.79}$ | $0.26_{\pm 0.20}$ |

Table 35: **Label Flip Attack on CIFAR-10(VGG16-BN)**
$\uparrow$: the goal is to have as high of a value as possible, $\downarrow$: the goal is to have as low of a value as possible. The top row presents the values of $\mathcal{M}^{(\theta^*)}$ on these metrics. These results show that RELOAD outperforms all baselines on NA, and performs competitively on OA and TA on which it is outperformed by GAR and FT respectively. RELOAD incurs a higher computational cost than most baselines, but is faster than FT, CF-$k$, and EU-$k$.

| Method | NA ($\uparrow$) | OA ($\uparrow$) | TA ($\uparrow$) | Cost ($\downarrow$) |
|---|---|---|---|---|
| Original | $91.06_{\pm 1.92}$ | $99.99_{\pm 0.00}$ | $87.31_{\pm 1.75}$ | N/A |
| Retrain | $99.99_{\pm 0.00}$ | $91.06_{\pm 1.92}$ | $95.45_{\pm 0.00}$ | $1.00_{\pm 0.00}$ |
| GAR | $91.05_{\pm 1.91}$ | $\mathbf{96.81_{\pm 2.24}}$ | $87.24_{\pm 1.78}$ | $0.08_{\pm 0.01}$ |
| GRDA | $25.10_{\pm 26.02}$ | $21.14_{\pm 25.56}$ | $23.43_{\pm 24.03}$ | $0.05_{\pm 0.00}$ |
| FT | $\mathbf{99.99_{\pm 0.00}}$ | $91.06_{\pm 1.92}$ | $\mathbf{95.39_{\pm 0.15}}$ | $0.41_{\pm 0.04}$ |
| SSD | $70.78_{\pm 21.90}$ | $72.70_{\pm 24.06}$ | $68.50_{\pm 20.77}$ | $\mathbf{0.01_{\pm 0.00}}$ |
| SCRUB | $6.76_{\pm 0.00}$ | $6.76_{\pm 0.00}$ | $6.71_{\pm 0.00}$ | $0.05_{\pm 0.00}$ |
| CF-$k$ | $92.56_{\pm 1.37}$ | $95.52_{\pm 6.50}$ | $88.77_{\pm 1.39}$ | $0.36_{\pm 0.01}$ |
| EU-$k$ | $92.48_{\pm 1.37}$ | $95.12_{\pm 6.31}$ | $88.58_{\pm 1.33}$ | $0.36_{\pm 0.01}$ |
| RELOAD | $99.42_{\pm 0.34}$ | $90.02_{\pm 1.47}$ | $94.75_{\pm 0.89}$ | $0.27_{\pm 0.13}$ |

Table 36: **Label Flip Attack on SVHN (VGG16-BN)**
$\uparrow$: the goal is to have as high of a value as possible, $\downarrow$: the goal is to have as low of a value as possible. The top row presents the values of $\mathcal{M}^{(\theta^*)}$ on these metrics. These results show that RELOAD is outperformed by other baselines on all metrics by a narrow margin and performs competitively across all metrics. RELOAD incurs a higher computational cost than most baselines, but is faster than FT, CF-$k$, and EU-$k$.