# OpenReview forum: "Blind Unlearning: Unlearning Without a Forget Set"
_ICLR.cc/2025/Conference — Submitted to ICLR 2025_

### Official Review · Reviewer_1dea · 2024-10-28

**Soundness:** 2
**Presentation:** 3
**Contribution:** 1
**Rating:** 3
**Confidence:** 4

**Summary:**

The paper introduces the concept of blind unlearning, which involves unlearning without explicit access to the forget set. The authors propose the RELOAD method, which utilizes gradient-based techniques and sparsity to achieve this form of unlearning.

**Strengths:**

The paper is well-written.

**Weaknesses:**

- The paper presents a setting where machine unlearning occurs without access to the forget dataset, which is weaker compared to zero-shot unlearning [1], where unlearning is achieved without access to either the forget or remaining datasets.

- The proposed setting is problematic. While the paper claims that blind unlearning does not access the forget dataset, it still allows access to the gradients of the entire dataset, including both the remaining and forget data. This is unreasonable because, to ensure data privacy, we typically use privacy-preserving mechanisms like DP-SGD. Gradients can leak data information, as attackers could exploit gradients to determine whether a specific data point was part of the training set. By analyzing how the gradient changes when a particular data point is used, adversaries can infer whether or not that point was included in training.

- Step 2 in Figure 1 is confusing. If you have computed all gradients on the retained dataset, why not simply retrain the model? Fine-tuning and gradient ascent methods are typically used for efficient unlearning, but this approach seems to overlook that option.

- The estimation of gradients on the forget dataset is not reasonable. The equations (5-7) fail when applied to average or mini-batch gradients, which makes the method problematic in practice.

- Several evaluation metrics, such as NSKL and FSKL, are unclear. The goal of unlearning is to forget the targeted data while retaining the performance of a model trained from scratch. Evaluating the model based on its output and data seems unnecessary and could detract from the focus on model performance.

References: [1] Chundawat, Vikram S., et al. "Zero-shot machine unlearning." IEEE Transactions on Information Forensics and Security 18 (2023): 2345-2354.

**Questions:**

I do not have specific question at the moment.

---

> ### Author Response · Authors · 2024-11-22
> **Response to Reviewer 1dea**
>
> We thank reviewer 1dea for their thoughtful comments. We’re encouraged that this reviewer found our paper to be well-written and structured.
>
> Below, we provide answers to their questions and remarks.
>
> > The paper presents a setting where machine unlearning occurs without access to the forget dataset, which is weaker compared to zero-shot unlearning [1], where unlearning is achieved without access to either the forget or remaining datasets.
>
> We agree with the reviewer that Zero-shot unlearning is a stronger setting than Blind unlearning. However, prior work which achieves zero-shot unlearning does so only with unlearning classes of data, which is a weaker setting than our setting of forgetting any subset of the training data. It is also a setting that enjoys significantly less broad applicability with regards to the motivating examples found in the general response to all reviewers.
>
> > The proposed setting is problematic. While the paper claims that blind unlearning does not access the forget dataset, it still allows access to the gradients of the entire dataset, including both the remaining and forget data. This is unreasonable because, to ensure data privacy, we typically use privacy-preserving mechanisms like DP-SGD. Gradients can leak data information, as attackers could exploit gradients to determine whether a specific data point was part of the training set. By analyzing how the gradient changes when a particular data point is used, adversaries can infer whether or not that point was included in training.
>
> Our theory shows that, given only the accumulated gradients of the loss with respect to each parameter, one cannot perfectly recover the instances associated with those gradients without further information (e.g., about the size of $D_{forget}$. Furthermore, by establishing and motivating the setting of blind unlearning, our work lays the groundwork for settings with stronger guarantees on forget-set recoverability. We hope that future work can explore methods for blind unlearning with guaranteed non-recoverability, and our present work serves as a stepping stone toward these contributions. On balance, given that the vast majority of the unlearning literature requires the raw forget-set, $D_{forget}$, a method that only requires access to the accumulated gradients seems to represent a significant step forward.
>
> > Step 2 in Figure 1 is confusing. If you have computed all gradients on the retained dataset, why not simply retrain the model? Fine-tuning and gradient ascent methods are typically used for efficient unlearning, but this approach seems to overlook that option.
>
> We believe that this is a misunderstanding. The gradients on the retained dataset are computed only once in Step 2, and only a single ascent step is applied. Retraining the model would require repeated computation of these gradients – hence why our results demonstrate that Reload is significantly more computationally efficient than fine-tuning or gradient ascent.
>
> > The estimation of gradients on the forget dataset is not reasonable. The equations (5-7) fail when applied to average or mini-batch gradients, which makes the method problematic in practice.
>
> We assume that the gradients cached at the end of training are full-batch and accumulated. Even when mini-batch gradients are used, these can be summed up and cached for the same effect.
>
> > Several evaluation metrics, such as NSKL and FSKL, are unclear. The goal of unlearning is to forget the targeted data while retaining the performance of a model trained from scratch. Evaluating the model based on its output and data seems unnecessary and could detract from the focus on model performance.
>
> One of the goals of unlearning is to produce a model that is a close approximation of the naively retrained one. NSKL and FSKL measure this by quantifying the dissimilarity between the outputs of the unlearned model and the retrained model on the same data.
>
> We hope this response has addressed the reviewer’s concerns, and we would be glad to continue the discussion if there are any remaining questions or comments. We thank the reviewer again for their attentive, detailed feedback.

---

> > ### Comment · Reviewer_1dea · 2024-11-26
> >
> > Thank you for the author's response. Below are my replies:
> >
> > 1. Regarding Response 1: The distinction between zero-shot and blind settings is clear; however, I believe the blind setting is comparatively weaker.
> >
> > 2. Regarding Response 2: If we have the sum of gradients for a dataset, e.g., $\sum_{i \in S} \nabla L\left(w, x_i\right)$, it is possible to infer whether $x_j$ is part of the dataset by computing $\\sum\_{j \\in S \\backslash\\left\\{x\_i\\right\\}} \\nabla L\\left(w, x\_i\\right)$.
> >
> > 3. Regarding Response 3: In Figure 1.(5), you start from new parameters and re-optimize. Could you clarify how the optimization is conducted? Additionally, while the proposed method claims computational efficiency, it seems to require significant memory resources.
> >
> > 4. Regarding Response 4: When using batch gradients, the scalars $1 / b$ and $1 / n$ are needed. Could you explain how the equality is derived under these conditions?
> >
> > 5. Regarding Response 5: The evaluation metric is now clear to me and should be discussed in the paper.

---

> ### Author Response · Authors · 2024-12-01
>
> We thank the reviewer for their response.
>
> 1. The blind setting is comparatively weaker, though considering zero-shot unlearning has not been achieved for unlearning random samples, blind unlearning can be considered another step towards zero-shot unlearning.
> 2. If we also have access to the ground truth of $\nabla \mathcal{L} (w, x_j)$ then yes. However, this assumes that we know that only a single item was removed from $\mathcal{S}$.
> 3. In this figure we start from the parameters produced after the ascent step and re-initialisation step. The optimization is conducted via whichever optimization process was originally used for training. The reviewer is correct that this will require significant memory resources, but will take far less time than retraining from scratch.
> 4. We take the assumption that these gradients are not averaged over the batch but simply accumulated, which is how the equality is derived.
> 5. The metrics are discussed in the paper in section 4.2 briefly and then more so in Table 6 in Appendix A.3

---

### Official Review · Reviewer_ST8s · 2024-11-02

**Soundness:** 2
**Presentation:** 2
**Contribution:** 1
**Rating:** 3
**Confidence:** 3

**Summary:**

This paper targets reducing the usage of forgetting data in machine unlearning and proposes a gradient ascent approach together with parameter selection. Due to the absence of forgetting data, the proposed method estimates the gradient of forgetting data by using the cached gradient on all training data and the gradient of the remaining data. Then, this paper conducts experiments on SVHN, Cfiar10 and Cifar100 datasets to prove the effectiveness of the proposed method.

**Strengths:**

1. Reducing the usage of forgetting data is an interesting problem for machine unlearning.
2. The technique and experiment part of this paper is easy to understand.

**Weaknesses:**

1. The motivation of this paper is to reduce the retaining of user data for unlearning. This paper only focuses on reducing the usage of forgetting data and requires the usage of remaining data. However, the size of the forgotten data is usually far smaller than the remaining data, and this also motivates other works that reduce the usage of the remaining data during unlearning [1,2,3]. Compared with such works, the proposed method required more training data to be preserved for unlearning. In addition, some methods that require both the forgetting data (10% in this paper's experiment setting ) and a subset of remaining data during unlearning (10% in this paper's experiment setting) only require 20% training to realise unlearning. However, the proposed method still requires another 90% of the remaining data during unlearning. Therefore, the proposed method does not match the motivation of reducing the data usage in unlearning
2. One previous work discussed reducing the usage of forgetting data, but this paper does not mention it in related works or compare it in experiments [4]. Other works have applied gradient-based input saliency maps for unlearning, and this paper does not mention or compare them [5,6].
3. This paper does not contain ablation studies to prove the effectiveness of the parameter selection component and different $\eta_p$ and $\alpha$.
4. This paper only conducts experiments on 10% random sample unlearning and 100 samples in class unlearning. More different experiment settings are required.
5. Some typos exist in the paper, for example, in line 147, and some notations are unclear (see questions).

[1] Chen, Min, et al. "Boundary unlearning: Rapid forgetting of deep networks via shifting the decision boundary." Proceedings of the IEEE/CVF Conference on Computer Vision and Pattern Recognition. 2023.

[2] Chundawat, Vikram S., et al. "Zero-shot machine unlearning." IEEE Transactions on Information Forensics and Security 18 (2023): 2345-2354.

[3] Zhang, Chenhao, et al. "GENIU: A Restricted Data Access Unlearning for Imbalanced Data." arXiv preprint arXiv:2406.07885 (2024).

[4] Tarun, Ayush K., et al. "Fast yet effective machine unlearning." IEEE Transactions on Neural Networks and Learning Systems (2023).

[5] Fan, Chongyu, et al. "SalUn: Empowering Machine Unlearning via Gradient-Based Weight Saliency in Both Image Classification and Generation." International Conference on Learning Representations. 2024.

[6] Huang, Zhehao, et al. "Unified Gradient-Based Machine Unlearning with Remain Geometry Enhancement." arXiv preprint arXiv:2409.19732 (2024).

**Questions:**

1. In experiments, why does a huge gap exist between the retrained model and $\mathcal{M}^{(\theta^{\sim})}$ under the current evaluations? What is the differences between retrained model and $\mathcal{M}^{(\theta^{\sim})}$.
2. In eq 8, does $\nabla_{\theta_k}\mathcal{L}(D)$ stand for the gradient of accumulated loss or averaged loss on $D$?
3. How to decide $\eta_p$ and $\alpha$ in the proposed method?
4. Minor question: regarding reducing the usage of forgetting data, how could users propose unlearning requests if they cannot point out that the forgetting data should be removed?

---

> ### Author Response · Authors · 2024-11-22
> **Response to Reviewer ST8s**
>
> We thank reviewer ST8s for their thoughtful comments. We’re encouraged that this reviewer found our setting to be an interesting problem and that they appreciated the algorithm and our experiments.
>
> Below, we provide answers to their questions and remarks.
>
> > The motivation of this paper is to reduce the retaining of user data for unlearning. This paper only focuses on reducing the usage of forgetting data and requires the usage of remaining data. However, the size of the forgotten data is usually far smaller than the remaining data, and this also motivates other works that reduce the usage of the remaining data during unlearning [1,2,3]. Compared with such works, the proposed method required more training data to be preserved for unlearning. In addition, some methods that require both the forgetting data (10% in this paper's experiment setting ) and a subset of remaining data during unlearning (10% in this paper's experiment setting) only require 20% training to realise unlearning. However, the proposed method still requires another 90% of the remaining data during unlearning. Therefore, the proposed method does not match the motivation of reducing the data usage in unlearning
>
> We believe this is a slight misunderstanding. The motivation of this paper is to reduce the retaining of the user data once that data has been requested for deletion. Additionally, the assignment of 10% of the training data to the forget set is simply a single setting that this can apply to, we now include the setting of forgetting 30% of the training data. We do agree with the reviewer that an interesting extension to this work would be using less of the retained user data to fine-tune the model after re-initialisation.
>
> > One previous work discussed reducing the usage of forgetting data, but this paper does not mention it in related works or compare it in experiments [4]. Other works have applied gradient-based input saliency maps for unlearning, and this paper does not mention or compare them [5,6].
>
> We thank the reviewer for bringing [4] to our attention. We have included it in the discussion. This work operates solely within the realm of class unlearning, which is a weaker and less useful setting of unlearning that we consider. [5] is discussed in depth in the paper, and we have included it in comparisons for unlearning now. [6] is a work that was released a day before the ICLR deadline, and as such was not brought to our attention in time nor would it have been feasible to include it at such a late stage.
>
> > This paper does not contain ablation studies to prove the effectiveness of the parameter selection component and different eta and alpha
>
> The reviewer raises a valid concern regarding the hyperparameter ablations, and we have included these in the appendix.
>
> > This paper only conducts experiments on 10% random sample unlearning and 100 samples in class unlearning. More different experiment settings are required.
>
> We have included 30% random sample unlearning in the appendix section.
>
> > In experiments, why does a huge gap exist between the retrained model and unlearned under the current evaluations? What is the differences between retrained model and unlearned.
>
> We believe that this is a notational misunderstanding. The retrained model is the gold-standard and is the same as $\mathcal{M}^{\theta
>  \sim}$.
>
> > In eq 8, does nabla stand for the gradient of accumulated loss or averaged loss on D?
>
>
> Accumulated.
>
> > How to decide eta and alpha in the proposed method?
>
> We decide $\eta$ and $\alpha$ using a Bayesian hyperparameter sweep.
>
> We hope this response has addressed the reviewer’s concerns, and we would be glad to continue the discussion if there are any remaining questions or comments. We thank the reviewer again for their attentive, detailed feedback.

---

> > ### Comment · Reviewer_ST8s · 2024-11-29
> > **Response to author replies**
> >
> > Thanks for your replies. My question about
> > But I still have the following concerns:
> > 1. Regarding motivation, as the author claims, this paper aims to reduce the retaining of user data for unlearning. In your experiments, you use 10%, 100, and 30% training samples for forgetting data. It is obvious that the remaining data size is much larger than the forgetting data. Thus, how can we guarantee that the proposed method uses less training data than previous work, which uses both the forgetting data and the same size of the remaining data subset as the forgetting data?
> > 2. Although [4] can only work on class unlearning, in this paper's experiments, class unlearning is involved. Thus, compared with [4] with a fair setting is essential.
> > 3. Regarding the last question, how could users propose unlearning requests if they cannot point out that the forgetting data should be removed? Are there any practical scenarios where the proposed setting happens, other than class unlearning?

---

### Official Review · Reviewer_x8sa · 2024-11-02

**Soundness:** 3
**Presentation:** 3
**Contribution:** 2
**Rating:** 6
**Confidence:** 4

**Summary:**

This paper proposed RELOAD framework that aims to achieve unlearning through the following steps:

Step (1-3): Compute the gradient of the  forget set of the last round (by subtracting the gradient of the remaining data from the retained full gradient) to perform a single step of gradient ascent.

Step (4): Building on previous unlearning method that assess weight importance, the author calculates the importance of the weights and reinitializes those deemed non-important.

Step (5): Fine-tune on the remaining data.

The author empirically demonstrates that the proposed method, which does not require access to the forget dataset, outperforms existing algorithms that necessitate access to the entire dataset

**Strengths:**

1. The author's method does not require access to the remaining dataset and may represent a significant algorithmic contribution.

1. The author proposes a new method for calculating knowledge values (weight importance), which involves computing the ratio of the L2 norm of the forget dataset to the L2 norm of the gradient of the entire dataset on a well-optimized model. The L2 norm of the forget dataset is obtained by subtracting the gradient of the remaining data from the gradient of the entire dataset.

2. The paper provides empirical results for different datasets and models and the numerical results are impressive because they perform better than previous algorithms that required access to all the data, and the detailed description of the metrics provided by the author in the appendix is well-written.

**Weaknesses:**

1. Although the author's method does not require access to the forget dataset, calculating $\nabla_\theta \mathcal{L}(\mathcal{D} _ {\text{forget}}) = \nabla _ \theta \mathcal{L}(\mathcal{D}) - \nabla_\theta \mathcal{L}(\mathcal{D} \backslash D _ {\text{forget}})$  in step (1-3) requires retaining all datasets to compute $\nabla _ \theta \mathcal{L}(\mathcal{D} \backslash D _ {\text{forget}})$ since we do not know which data might need to be forgotten. I suggest the author describe potential application scenarios more in the introduction.

2. Retaining $\nabla_\theta \mathcal{L}(\mathcal{D})$  in step (1-3) is risky because prior work [B] has demonstrated that, for fully connected layers (such as the softmax mentioned by the authors), the input can theoretically always be inferred from the gradients, regardless of the layer's position. Experiments of [B] have also shown that images can be reconstructed from gradients. Therefore, when considering the retention of $\nabla_\theta \mathcal{L}(\mathcal{D})$, the authors should take additional measures to prevent unintended privacy leakage.

3. The author evaluates the importance of weights in step (4), drawing inspiration from prior unlearning work [A]. However, the paper lacks a comparison with existing methods for assessing weight importance. It’s crucial to clarify the differences between the gradient norm of the forget dataset used in previous studies and the ratio employed by the author as knowledge values. Simply stating in the appendix that their method outperforms other weight importance methods is insufficient; the author should at least report experimental results to justify the choice of knowledge values and help readers understand the contributions. Furthermore, the baseline analysis appears to overlook a comparison with the significant work [A], which should be addressed for a more comprehensive evaluation.

4. In the title of this paper and contribution, the author mentions that their method does not require access to the "forget dataset." However, this is not a new problem (motivation). There are several methods, which the author has not cited (e.g., [C-F] or references listed in [G]), that also do not rely on the "forget dataset." I suggest that the author explicitly address the relevance of these existing methods when discussing the uniqueness of their approach and explain the differences and connections between their research and these methods. This would help readers better understand the author's contributions and the work done on the foundation of existing research.

5. The author claims that RELOAD framework outperform existing algorithms that require access to the entire dataset, but I am concerned about the lack of either theoretical guarantees and empirical intuition behind the finds.   The paper lacks many reproducibility details regarding the hyper-parameters used (their method and the comparison methods) and the fine-tuning details. There's no way to verify the correctness of the result in the paper or to reproduce any of the results if the paper is accepted.  At a minimum, the author should provide insights to explain this phenomenon observed in the experiments, such as which key steps were missing in previous works that led to results inferior to the algorithm proposed in this paper, or how the author's framework helps enhance performance, reveal potential issues, or optimize the implementation of the algorithm. This would help readers better understand the contributions and practical significance of the research. Unfortunately, I found no explanations, and the links provided by the author contain no verifiable procedures.  I suggest that the author provide some explanations to clarify the phenomena observed in the experiments, or offer verifiable procedures to enhance the paper's persuasive power. This would help improve the credibility of the research and better showcase its algorithmic contributions.

**Questions:**

1. I am curious about the contribution of  step (1-3) one-step gradient ascent to the overall framework. Is step (4) (5) sufficient to satisfy unlearning? I did not see the author provide relevant explanations for step (1-3), which makes it difficult to detail the necessity of step (1-3). I suggest that the authors highlight the necessity of step (1-3) in the unlearning framework through some ablation experiments; otherwise, (4) (5) seem to be mere improvements based on previous methods.

2. Equations (4) to (7) seem unnecessary, as they occupy a significant amount of space without providing any useful information. The meaning could be effectively conveyed with just $\nabla_\theta \mathcal{L}(\mathcal{D}_{\text{forget}}) = \nabla_\theta \mathcal{L}(\mathcal{D}) - \nabla_\theta \mathcal{L}(\mathcal{D}_{\text{new}})$.

3. Blind unlearning is an interesting concept; however, given that prior work has achieved zero-shot unlearning without requiring access to any training data for the unlearning process, it is puzzling that this paper still necessitates access to the remaining dataset. The term "blind unlearning" could lead to confusion. A title like "Partially Blinded Unlearning," indicating that only a portion of the data is accessible (such as the remaining data or part of the forget dataset), might be more appropriate.

[A] SalUn: Empowering Machine Unlearning via Gradient-Based Weight Saliency in Both Image Classification and Generation. Fan, Chongyu, et al. ICLR, 2024.

[B] Inverting gradients-how easy is it to break privacy in federated learning? Geiping, Jonas, et al. NeurIPS, 2020.

[C] Eternal sunshine of the spotless net: Selective forgetting in deep networks. Golatkar et al. CVPR, 2020.

[D] Fast yet effective machine unlearning.Tarun, Ayush K., et al. TNNLS, 2023.

[E] Deep Regression Unlearning. Ayush Kumar Tarun, et al. ICML, 2023

[F] Deep Unlearning: Fast and Efficient Gradient-free Class Forgetting. Sangamesh Kodge, et al. TMLR, 2024.

[G] LLM Unlearning via Loss Adjustment with Only Forget Data

[H] Zero-Shot Machine Unlearning. Vikram S Chundawat, et al. TMLR, 2024.

---

> ### Author Response · Authors · 2024-11-22
> **Response to Reviewer x8sa**
>
> We thank reviewer x8sa for their thoughtful comments. We’re encouraged that this reviewer found our work a potential significant algorithmic contribution and that they appreciated the breadth of our comparisons against baseline algorithms.
>
> Below, we provide answers to their questions and remarks.
>
> > Although the author's method does not require access to the forget dataset, calculating $\nabla_\theta \mathcal{L}(\mathcal{D}{\text{forget}}) = \nabla\theta \mathcal{L}(\mathcal{D}) - \nabla_\theta \mathcal{L}(\mathcal{D} \backslash D_{\text{forget}})\nabla_\theta \mathcal{L}(\mathcal{D} \backslash D_{\text{forget}})$ since we do not know which data might need to be forgotten. I suggest the author describe potential application scenarios more in the introduction.
>
> We thank the reviewer for this concern. We have addressed further potential application scenarios in the response to all reviewers. We believe this is a slight misunderstanding. Reload assumes that the practitioner training the model retains the full batch gradients from the end of training, thus meaning that all the training data (including the “forget set”) need not be retained for unlearning.
>
> > Retaining  in step (1-3) is risky because prior work [B] has demonstrated that, for fully connected layers (such as the softmax mentioned by the authors), the input can theoretically always be inferred from the gradients, regardless of the layer's position. Experiments of [B] have also shown that images can be reconstructed from gradients. Therefore, when considering the retention of , the authors should take additional measures to prevent unintended privacy leakage.
>
> The reviewer raises a valid concern and we thank you for that. Experiments of [B] mentioned have shown that parts of images can be reconstructed from gradients, however, the work mentions that most of the images are unrecognizable and the small portion of the batch that have the most reconstruction, are still perturbed. Additionally, in order to reconstruct this batch, [B] requires knowing the number of images in the batch to produce a prior, which is not a quantity that is known in Reload. We believe this is a valid concern, especially as work progresses in this area, and note that it would be an important future direction to study defenses against better methods of hiding data information in gradients.
>
> > The author evaluates the importance of weights in step (4), drawing inspiration from prior unlearning work [A]. However, the paper lacks a comparison with existing methods for assessing weight importance. It’s crucial to clarify the differences between the gradient norm of the forget dataset used in previous studies and the ratio employed by the author as knowledge values. Simply stating in the appendix that their method outperforms other weight importance methods is insufficient; the author should at least report experimental results to justify the choice of knowledge values and help readers understand the contributions. Furthermore, the baseline analysis appears to overlook a comparison with the significant work [A], which should be addressed for a more comprehensive evaluation.
>
> The reviewer raises a valid point with the lack of comparison with regards to the weight importances calculated in [A]. A deeper comparison has been added to the appendix in order to justify the formulation of the knowledge values in comparison to [A]. We agree that comparison with [A] should be done, and these results have been added to the paper.

---

> > ### Author Response · Authors · 2024-11-22
> > **Response to Reviewer x8sa (continued)**
> >
> > > In the title of this paper and contribution, the author mentions that their method does not require access to the "forget dataset." However, this is not a new problem (motivation). There are several methods, which the author has not cited (e.g., [C-F] or references listed in [G]), that also do not rely on the "forget dataset." I suggest that the author explicitly address the relevance of these existing methods when discussing the uniqueness of their approach and explain the differences and connections between their research and these methods. This would help readers better understand the author's contributions and the work done on the foundation of existing research.
> >
> > We thank the reviewer for bringing these to our attention. We have included [C] in discussion as a blind unlearning algorithm. The work introduced in [C] can be considered “blind unlearning” but is a limited method as it is only valid in class or subclass unlearning, however, we have added comparisons to the Fisher forgetting method introduced in the work to our evaluations for unlearning, and are working on testing it for remedial learning. [D] requires knowing which class is being unlearned and thus cannot be considered “blind unlearning” as this case falls outside of the setting. Additionally, [E] and [F] utilize some or all of the forget set in their optimization loops and thus also fall outside of the setting of “blind unlearning”. Unfortunately, we cannot find the references listed in [G] that the reviewer is referring to. It would be helpful if the reviewer could point to those directly. Nevertheless we thank the reviewer for raising these, as they are valid concerns regarding the novelty of our setting.
> >
> > >The author claims that RELOAD framework outperform existing algorithms that require access to the entire dataset, but I am concerned about the lack of either theoretical guarantees and empirical intuition behind the finds. The paper lacks many reproducibility details regarding the hyper-parameters used (their method and the comparison methods) and the fine-tuning details. There's no way to verify the correctness of the result in the paper or to reproduce any of the results if the paper is accepted. At a minimum, the author should provide insights to explain this phenomenon observed in the experiments, such as which key steps were missing in previous works that led to results inferior to the algorithm proposed in this paper, or how the author's framework helps enhance performance, reveal potential issues, or optimize the implementation of the algorithm. This would help readers better understand the contributions and practical significance of the research. Unfortunately, I found no explanations, and the links provided by the author contain no verifiable procedures. I suggest that the author provide some explanations to clarify the phenomena observed in the experiments, or offer verifiable procedures to enhance the paper's persuasive power. This would help improve the credibility of the research and better showcase its algorithmic contributions.
> >
> > Our linked repository contains fully reproducible code for our experiments.
> >
> > > I am curious about the contribution of step (1-3) one-step gradient ascent to the overall framework. Is step (4) (5) sufficient to satisfy unlearning? I did not see the author provide relevant explanations for step (1-3), which makes it difficult to detail the necessity of step (1-3). I suggest that the authors highlight the necessity of step (1-3) in the unlearning framework through some ablation experiments; otherwise, (4) (5) seem to be mere improvements based on previous methods.
> >
> > Step 4-5 are sufficient to satisfy unlearning an entire class from the training data. We find empirically that it isn’t sufficient when deleting random samples from the training data, as there may be little to no correlation or underlying structure to these samples. As such, Steps 1-3 are used to remove top-level information about the “forget set” from the parameters that we do not reinitialize.

---

> > > ### Author Response · Authors · 2024-11-22
> > > **Response to Reviewer x8sa (continued)**
> > >
> > > > Equations (4) to (7) seem unnecessary, as they occupy a significant amount of space without providing any useful information. The meaning could be effectively conveyed with just $\nabla_\theta \mathcal{L}(\mathcal{D}{\text{forget}}) = \nabla\theta \mathcal{L}(\mathcal{D}) - \nabla_\theta \mathcal{L}(\mathcal{D}_{\text{new}})$.
> > >
> > > The contribution of Equations (4) through (7) does not lie in mathematical novelty, but rather in the observation that linearity of differentiation allows one to treat the difference between the (cached) gradients of the loss with respect to $\mathcal{D}$ and the gradients of the loss with respect to $\mathcal{D}_{new}$ as the direction of movement in parameter space needed to unlearn the influence of the unobserved $\mathcal{D}_{forget}$. The reviewer is correct that the core of this step is mathematically straightforward. However, our results highlight its empirical utility in allowing Reload to outperform state-of-the-art baseline benchmarks. We would argue that simple-but-effective ideas hold outsized value for the machine learning community due to their ease of implementation.
> > >
> > > >Blind unlearning is an interesting concept; however, given that prior work has achieved zero-shot unlearning without requiring access to any training data for the unlearning process, it is puzzling that this paper still necessitates access to the remaining dataset. The term "blind unlearning" could lead to confusion. A title like "Partially Blinded Unlearning," indicating that only a portion of the data is accessible (such as the remaining data or part of the forget dataset), might be more appropriate.
> > >
> > > We agree with the reviewer that Zero-shot unlearning is a stronger setting than Blind unlearning. However, prior work which achieves zero-shot unlearning does so only with unlearning classes of data, which is a weaker setting than our setting of forgetting any subset of the training data. It is also a setting that enjoys significantly less broad applicability with regards to the motivating examples found in the general response to all reviewers.
> > >
> > > We hope this response has addressed the reviewer’s concerns, and we would be glad to continue the discussion if there are any remaining questions or comments. We thank the reviewer again for their attentive, detailed feedback.

---

> > > ### Comment · Reviewer_x8sa · 2024-11-24
> > >
> > > Thanks to the authors for their response. During the rebuttal period, the authors provided code to reproduce the results, which is appreciated. However, I still suggest that future versions include a more detailed description of the hyperparameter selection.
> > >
> > > The authors also provided a comparison with the related work [A]. While helpful, I am still somewhat unclear about why the authors' approach outperforms other methods that fully utilize the dataset in the experiments. Additional explanations or experiments in future versions (though not mandatory) could further highlight the contributions.
> > >
> > > I only have one question: For the equation $\nabla _ \theta \mathcal{L}(\mathcal{D} _ {\text{forget}}) = \nabla _ \theta \mathcal{L}(\mathcal{D}) - \nabla _ \theta \mathcal{L}(\mathcal{D} \backslash D _ {\text{forget}})$, where the practitioner training the model retains the full batch gradients $\nabla _ \theta \mathcal{L}(\mathcal{D})$ from the end of training, how should the gradient $\nabla_\theta \mathcal{L}(\mathcal{D} \backslash D _ {\text{forget}})$ be computed during the unlearning process?
> > >
> > > Aside from this, I have no further questions. Considering the comprehensive experiments and robust results, I will maintain my current rating.

---

> > > > ### Author Response · Authors · 2024-11-25
> > > >
> > > > We thank the reviewer for their response.
> > > >
> > > > As to their question, the mentioned gradient can be computed as this data is available to the practitioner as the retain set.

---

### Official Review · Reviewer_rRbE · 2024-11-04

**Soundness:** 2
**Presentation:** 1
**Contribution:** 2
**Rating:** 3
**Confidence:** 3

**Summary:**

This paper addresses a critical and well-motivated setting in recent machine unlearning research: unlearning without explicit access to the forget set, termed "blind unlearning" by the authors. The paper introduces an approximate unlearning method called RELOAD, which combines a single step of gradient ascent with a selective re-initialization procedure. Additionally, the RELOAD method can be extended to a remedial learning setting, which generalizes the classical unlearning problem and aims to efficiently update a previously trained model to accommodate an amended dataset.

**Strengths:**

S1. The problem of "unlearning without explicit access to the forget set" is well-motivated and has attracted significant attention.

S2. The "remedial learning" setting considered in this paper is interesting.

**Weaknesses:**

W1. The paper lacks structure and clarity in its writing. For example:
* To the best of my knowledge, "remedial learning" seems to be a new setting introduced in this paper. If this is not the case, please provide the necessary references upon its first mention. Regardless, the authors should provide sufficient background on this setting in the Abstract, Introduction, as well as Title. Failing to do so may lead to unnecessary difficulties and confusion for readers attempting to understand the task's objective from the beginning.
* In Line 211 of Section 3.1, the statement "Recall from the relationship between ∇θL(D) and ∇θL(Dnew) in Section 3.2, …." is confusing.
* In Line 213, it appears that "the numerator ∇θk L(Dforget)" refers to the numerator in Eq. (8) (line 267). Introducing a quantity before it is first presented in the text can be confusing for readers.
* Typo in line 183: "at the instanced of unlearning"
* Typo in line 209: "\sum_{i=1}^N"

W2. The theoretical novelty of the derivations from Eq. (4) to Eq. (8) is unclear. The result $\nabla_\theta L(D_f) = \nabla_\theta L(D) - \nabla_\theta L(D_retain)$ seems simple and straightforward. I would appreciate further clarification on the contribution of this part.

W3. Please correct me if I misunderstood. From line 8 in Algorithm 1, it appears that selective re-initialization is performed on each element in the parameter vector. Intuitively, this design could increase the computational burden. Please explain this point. Additionally, in line 9 of Algorithm 1, a hyper-parameter $\alpha$ is introduced as a threshold to determine whether a parameter should be initialized. Is this hyperparameter the same for all model parameters? Ideally, it should vary, and discussing a suitable strategy for selecting an appropriate $\alpha$ for different parameters is recommended.

**Questions:**

In addition to addressing W1-W3, the authors are also expected to answer the following simple questions:

Q1: In Line 241, what do you mean by “single-based” ascent update?

Q2: What’s the impact of the Laplace smoothing constant ε in Eq (8)? How should its value be chosen?

---

> ### Author Response · Authors · 2024-11-22
> **Response to Reviewer rRbE**
>
> We thank the reviewer for their thoughtful comments. We’re encouraged that this reviewer found our problem setting “well-motivated,” and that they enjoyed our extension to the setting of remedial learning.
>
> Below, we provide answers to their questions and remarks.
>
> > The paper lacks structure and clarity in its writing. For example: To the best of my knowledge, "remedial learning" seems to be a new setting introduced in this paper. If this is not the case, please provide the necessary references upon its first mention. Regardless, the authors should provide sufficient background on this setting in the Abstract, Introduction, as well as Title. Failing to do so may lead to unnecessary difficulties and confusion for readers attempting to understand the task's objective from the beginning.
>
> This understanding is correct. “Remedial learning” represents a new setting introduced in this paper. It comprises the scenario wherein the practitioner wants to unlearn the influence of certain aspects of $\mathcal{D}$ on the model, and to replace them with the corresponding aspects of $\mathcal{D}_{new}$
>
> (e.g., unlearning a spurious backdoor from $\mathcal{D}$, and replacing the influence of the backdoor with the components of $\mathcal{D}_{new}$ that do not contain the backdoor in question). We have edited the manuscript to clarify that remedial learning is a novel setting introduced in this work.
>
> We have fixed the other writing concerns raised by this reviewer in lines 211, 213, 183, and 209, and we sincerely thank them for their attentive eye to detail.
>
> > The theoretical novelty of the derivations from Eq. (4) to Eq. (8) is unclear. The result seems simple and straightforward. I would appreciate further clarification on the contribution of this part.
>
> The contribution of Equations (4) through (8) does not lie in mathematical novelty, but rather in the observation that linearity of differentiation allows one to treat the difference between the (cached) gradients of the loss with respect to $\mathcal{D}$ and the gradients of the loss with respect to $\mathcal{D}_{new}$
>
> as the direction of movement in parameter space needed to unlearn the influence of the unobserved $\mathcal{D}_{forget}$. The reviewer is correct that the core of this step is mathematically straightforward. However, our results highlight its empirical utility in allowing Reload to outperform state-of-the-art baseline benchmarks. We would argue that simple-but-effective ideas hold outsized value for the machine learning community due to their ease of implementation.
>
> > Please correct me if I misunderstood. From line 8 in Algorithm 1, it appears that selective re-initialization is performed on each element in the parameter vector. Intuitively, this design could increase the computational burden. Please explain this point. Additionally, in line 9 of Algorithm 1, a hyper-parameter  is introduced as a threshold to determine whether a parameter should be initialized. Is this hyperparameter the same for all model parameters? Ideally, it should vary, and discussing a suitable strategy for selecting an appropriate  for different parameters is recommended.
> Knowledge values are computed for all parameters. However, only parameters with knowledge values less than that $\alpha$-quantile of the empirical distribution of knowledge values over all parameters are selectively re-initialized. We’ve amended this line of the algorithm to use a more conventional if-block to avoid confusion.
>
> > A hyper-parameter is introduced as a threshold to determine whether a parameter should be initialized. Is this hyperparameter the same for all model parameters? Ideally, it should vary, and discussing a suitable strategy for selecting an appropriate for different parameters is recommended.
>
> The hyperparameter threshold $\alpha$ is the same for all model parameters. It cannot vary
>
> > Q1: In Line 241, what do you mean by “single-based” ascent update?
>
> This is a typo. We’ve corrected it to read “single gradient ascent update step.”
>
> > What’s the impact of the Laplace smoothing constant ε in Eq (8)? How should its value be chosen?
>
> The Laplace smoothing constant in Equation 8 is to prevent division by zero. Additionally, it allows the equation to maintain measuring relative importance when the numerator is 0, avoiding assigning the same KV to all parameters with numerator 0 and denominator non-zero  Its value need only be small – we use 1e-10 in our experiments.
>
> We hope this response has addressed the reviewer’s concerns, and we would be glad to continue the discussion if there are any remaining questions or comments. We thank the reviewer again for their attentive, detailed feedback.

---

### Official Review · Reviewer_f5or · 2024-11-05

**Soundness:** 1
**Presentation:** 2
**Contribution:** 2
**Rating:** 3
**Confidence:** 4

**Summary:**

The paper proposes the setting of Blind unlearning. In short this setting deals with cases when the unlearning algorithm does not have access to the forget set but to some other information. The proposed algorithm in the paper requires access to the retain set, the trained model, and multiple gradient checkpoints. In essence the paper does gradient ascent on the forget set where the gradients on the forget set are computed by taking the difference of the gradients on the full set (checkpoints) and gradients on the retain set (can be computed ). The paper also proposes the setting of reemdial unlearning which also requires access to the "clean" dataset that can remedy the unlearned dataset.

**Strengths:**

The paper looks at the problem of unlearning without full access to the forget set. The algorithmic techniques combine few different ideas already present in the literature in a clean way - checkpointing, gradient ascent, and sparsity.

The concept of blind unlearning, where the unlearner does not have full access to the forget set could be an interesting setting to consider.

I also appreciated that the authors decided to evaluate on several metrics and across a wide array of baseline algorithms for unlearning.

**Weaknesses:**

There are three main weakness of the paper which I believe makes the paper unsuitable for publishing at its current stage.

1. __Theoretical claims__ There are multiple issues with the theoretical claims of the paper and unless I have misunderstood them (pleasr correct me if I am wrong), I dont think they are fixable without significantly altering the contributions of the paper.
    * The definition of _recoverability_ says is that if $f$ is injective then all datasets $D$ are recoverable. This does not mean that if $f$ is not injective then no D is recoverable. it may be possible that some Ds are recoverable.
    * The paper looks at _exact recoverability_ which may be not only be unrealistic in practice but also an unnecessarily high bar. it is often impossible to exactly recover training data points but recover them to a high degree. As such, there should be a notion of approx. recovery.
    * The theoretical intuition behind the algorithm is that the gradients on the forget set can be approximated by the difference of the gradients in the full dataset and the retain set. However, these are only true for the check point parameters save and it is difficult to argue (not done in the paper) why they should be similar after some gradient ascent steps are done.
2. _Lack of diversity of experimental results and comparisons in known baselines_ Previous works including pawelczyk et. al. and Goel et. al. have considered many relevant experimental baselines which this work must also look at.
    * For unlearning, consider the IC test in Goel et. al. and the targeted, indiscriminate, and gaussian data poisoning attacks in Pawelczyk et. al.
    * For remedial unlearning, it is easy enough to adapt the above test by using clean data points.
    * The experiments only use ResNet18 and VGG on CIFAR10 and CIFAR100. These are not sufficient to make generalisable claims especially when the paper is empirical in nature.
    In general these two papers should be discussed as relevant existing work.
3. My final relatively milder criticism is that I do not see the motivation for when this setting is realistic. When is it that the learner only has access to a large retain set (90% of the dataset), multiple gradient checkpoints during training, but not the forget set. This selectively requires the learner to not have the forget set but have all these other memory intensive data structures.



Pawelczyk, Martin, et al. "Machine unlearning fails to remove data poisoning attacks." arXiv preprint arXiv:2406.17216 (2024).
Goel, Shashwat, et al. "Corrective machine unlearning." arXiv preprint arXiv:2402.14015 (2024).

**Questions:**

1. SSD results look unrealistic. The accuracy seems to be 1% on CIFAR100 which is the trivial accuracy. Previous results with SSD have shown much better performance on clean test acuracy with SSD. It appears the hyper-param optimisation may not have been executed properly.
2. for MIA, what is the reported number ? Is it a fraction between 0 and 1 or a percentage between 0 and 100 (like the other columns). This number appears too low if its a percentage to be meaningful. Can the authors report AUC instead ?
3. Clarify the weaknesses above.

---

> ### Author Response · Authors · 2024-11-22
> **Response to Reviewer f5or**
>
> We thank reviewer f5or for their thoughtful comments. We’re encouraged that this reviewer found our problem setting “interesting to consider,” and that they appreciated the breadth of our comparisons against baseline algorithms.
>
> Below, we provide answers to their questions and remarks.
>
> ## Theoretical Claims
>
> > The definition of recoverability says is that if $f$ is injective then all datasets $D$ are recoverable. This does not mean that if $f$ is not injective then no $D$ is recoverable. It may be possible that some $D$s are recoverable.
> > The paper looks at exact recoverability which may be not only be unrealistic in practice but also an unnecessarily high bar. it is often impossible to exactly recover training data points but recover them to a high degree. As such, there should be a notion of approx. recovery.
>
> These two points are well-taken: they represent limitations of our present theory and setting and we will clarify this in the manuscript. However, by establishing and motivating the setting of blind unlearning, our work lays the groundwork for settings with stronger guarantees on forget-set recoverability. We hope that future work can explore methods for blind unlearning with guaranteed non-recoverability, and our present work serves as a stepping stone toward these contributions. On balance, given that the vast majority of the unlearning literature requires $D_{forget}$, even a method that only considers exact recoverability represents a valuable step forward.
>
> > The theoretical intuition behind the algorithm is that the gradients on the forget set can be approximated by the difference of the gradients in the full dataset and the retain set. However, these are only true for the check point parameters save and it is difficult to argue (not done in the paper) why they should be similar after some gradient ascent steps are done.
>
> We believe this represents a small misunderstanding. Reload requires only that the gradients from the last iteration of training be cached, and only employs them in a single ascent step (Figure 1; Step 3). The reviewer is correct that there is no reason to believe these gradients should be similar after multiple ascent steps, hence why Reload only performs one such weight update.
>
> ## Diversity of experimental results and comparisons in known baselines
> > For unlearning, consider the IC test in Goel et. al. and the targeted, indiscriminate, and gaussian data poisoning attacks in Pawelczyk et. al.
>
> The IC test in Goel et. al. deliberately mislabels some of the instances in $\mathcal{D}_{train}$, and then tests the ability of the unlearning procedure to unlearn mislabelled instances. Our paper includes remedial learning experiments in this setting using Reload in Appendix A.9. Furthermore, the targeted, indiscriminate, and gaussian data poisoning attacks in Pawelczyk et. al. is conceptually very similar to the backdoor correction experiments that feature in the main body (Section 4.3);. The key difference between the IC test and our setting in remedial learning is that while the IC test evaluates label misspecification, our experiments evaluate covariate misspecification. That Reload often outperforms baseline methods in both settings (and does so with significantly less computational resources) suggests that it fulfills the desired properties.
>
> > For remedial unlearning, it is easy enough to adapt the above test by using clean data points.
>
> It is correct that fine-tuning on clean data points is a valid means of performing remedial unlearning. However, our results suggest that Reload is a more efficient algorithm for remedial unlearning. Our findings in Table 3 demonstrate that Reload achieves a significantly higher accuracy than fine-tuning on a held-out test split of $\mathcal{D}_{new}$
>
>  (86.87 vs. 90.81), and on a modified held-out test split of  $\mathcal{D}_{new}$ with backdoors injected into each sample (86.50 vs. 90.51). More importantly, Reload does so at significantly lower cost – whereas fine-tuning requires 37% of the computational resources as from-scratch retraining, Reload requires only 8%.
>
> > The experiments only use ResNet18 and VGG on CIFAR10 and CIFAR100. These are not sufficient to make generalisable claims especially when the paper is empirical in nature. In general these two papers should be discussed as relevant existing work.
>
> Additional experiments on SVHN are presented in the supplementary materials. More importantly, the experiments provided in the paper study a wide range of unlearning and remedial learning scenarios, ranging from unlearning randomly-selected samples, to unlearning correlated samples, to removing shortcuts. If there are specific experiments this reviewer would like to see, we are open to suggestions and we can work to implement

---

> ### Author Response · Authors · 2024-11-22
> **Response to Reviewer f5or (continued)**
>
> ## Practicality of Setting
> > My final relatively milder criticism is that I do not see the motivation for when this setting is realistic. When is it that the learner only has access to a large retain set (90% of the dataset), multiple gradient checkpoints during training, but not the forget set. This selectively requires the learner to not have the forget set but have all these other memory intensive data structures.
> Please see the general response to all reviewers for a discussion of this concern.
>
> ## Questions
>
> >SSD results look unrealistic. The accuracy seems to be 1% on CIFAR100 which is the trivial accuracy. Previous results with SSD have shown much better performance on clean test acuracy with SSD. It appears the hyper-param optimisation may not have been executed properly.
>
> Our implementation was  lifted directly from the official reference implementation for SSD. Additionally it is important to note that SSD is not tested on randomly forgetting 10% of CIFAR-10 and is limited to deleting 100 samples - which could potentially lead to subpar performance.
>
> >for MIA, what is the reported number ? Is it a fraction between 0 and 1 or a percentage between 0 and 100 (like the other columns). This number appears too low if its a percentage to be meaningful. Can the authors report AUC instead?
>
> The reported number is a fraction between 0 and 1. We have included AUC in the 30% Forgetting experiments provided in Appendix A7.
>
> We hope this response has addressed the reviewer’s concerns, and we would be glad to continue the discussion if there are any remaining questions or comments. We thank the reviewer again for their detailed and thoughtful input on our work.

---

> > ### Comment · Reviewer_f5or · 2024-11-25
> >
> > I thank the authors for their response. I have read them and I decided to maintain my score. The mean reasons are that
> > * weakness 1 is still very important and as I mentioned and the authors agreed, this is not solvable.
> > * For weakness, 2 I would have liked to see the evaluations performed in Pawelczyk et. al. and Goel et. al. as these are perhaps some of the most related recent works with baselines.
> > * For weakness 3, I am still not convinced and would have liked a more to-the-point discussion regarding the various facets/requirements of blind unlearning.
> >
> > Nevertheless I thank the authors for clarifying several other aspects of my review like the metric for MIA (it seems the tables in the paper don't report AUC yet) and also the number of steps of gradient ascent.
> >
> > I believe with the above clarifications, the paper can make a good contribution to future conferences.

---

> > > ### Author Response · Authors · 2024-11-25
> > >
> > > We thank the reviewer for their response.
> > >
> > > The tables in Appendix A.7 for 30% Random Forgetting include AUC measurements at the moment. We are currently re-running all of the other experiments to include them as well. As to the number of steps of gradient ascent, RELOAD always uses only 1 step of gradient ascent. We apologise that this was not clear in our response.

---

### Author Response · Authors · 2024-11-22
**Response to All Reviewers**

We wish to thank all the reviewers for their careful and thoughtful comments.

We have made significant revisions to the manuscript to address reviewer questions and concerns. These changes are highlighted in blue.

**Setting and Motivation**. Several reviewers were concerned that the setting of blind unlearning – unlearning without explicit access to the “forget set” – may be unrealistic. Below, we present several applied scenarios to highlight the practicality of this setting.

1. **Pre-Trained Black-Box Models for AI Safety**. Assume that a practitioner is provided access to a pre-trained black-box image model, and that the analyst is tasked with removing knowledge of an undesirable topic (e.g., explicit imagery), from the model. Because the model is pre-trained, the practitioner does not have access to its training data, and so cannot construct a “forget set” for unlearning. Rather than retrain the model from scratch, we would propose that the practitioner use Reload, with $\mathcal{D}_{new}^*$ equal to a set of similar data that has been screened to not contain the undesirable topic in question. The model provider would only need to provide cached gradients from the last epoch of model training. Then, the Reload algorithm would allow the model to unlearn the undesirable topic efficiently without from-scratch retraining.

2. **On-Demand Data Deletion**. Imagine an organization serving a model trained on its user data, and that  a user wants his or her data deleted from the organization’s database and from any of the organization’s trained models. Using Reload, this organization has the flexibility to immediately delete this user’s data from the database, rather than retaining it for downstream unlearning. By permitting immediate deletion, Reload allows the organization to implement data management practices that better abide by the established best-practice Principle of Least Privilege (PLoP) [1,2], from the computer security literature.

**Broader Context**. We would also like to highlight that when the “forget set” is available, gradient caching for Reload is not required, because $\nabla_\theta \mathcal{L}( \mathcal{D})$ can be readily computed. In this setting, Reload still outperforms current state-of-the-art algorithms, which, in our experiments, were provided by the forget set. These results suggest that our method of serially applying an ascent step, targeted parameter reinitialization, and fine-tuning holds merit as an empirically effective approach.

[1] Needham, Roger M. "Protection systems and protection implementations." In Proceedings of the December 5-7, 1972, fall joint computer conference, part I, pp. 571-578. 1972.

[2] Schneider, F. B. (2004). Least privilege and more. In Computer Systems: Theory, Technology, and Applications (pp. 253-258). New York, NY: Springer New York.

---

### Meta-Review · Area_Chair_qH2w · 2024-12-17

**Metareview:**

This paper introduces a method for unlearning without a forget set. Unfortunately, the reviewers identified several significant issues, including that the definitions are problematic (e.g., exact unlearning is too strong and not even necessary), and the paper lacks comparison with appropriate baselines (e.g., Pawelczyk et. al. and Goel et. al.). Given these limitations, the paper is not ready for publication.

**Additional Comments On Reviewer Discussion:**

The reviewers identified several significant issues, including that the definitions are problematic (e.g., exact unlearning is too strong and not even necessary), and the paper lacks comparison with appropriate baselines (e.g., Pawelczyk et. al. and Goel et. al.). The authors were not able to address these concerns during rebuttal.

---

### Decision · Program_Chairs · 2025-01-22

Reject